# Response of seasonal soil freeze depth to climate change across China

Xiaoqing Peng[1], Tingjun Zhang[1][*], Oliver W. Frauenfeld[2], Kang Wang[3], Bin Cao[1], Xinyue Zhong[4], Hang Su[1], Cuicui Mu[1]

[1] Key Laboratory of Western China's Environmental Systems (Ministry of Education), College of Earth and Environmental Sciences, Lanzhou University, Lanzhou, 730000, China

[2] Department of Geography, Texas A&M University, College Station, TX 77843-3147, USA

[3] Institute of Arctic and Alpine Research, University of Colorado at Boulder, Boulder, CO 80309, USA

[4] Northwest Institute of Eco-Environment and Resources, Chinese Academy of Sciences, Lanzhou 730000, China

* *Correspondence to:* Tingjun Zhang (tjzhang@lzu.edu.cn)

**Abstract.** The response of seasonal soil freeze depth to climate change has repercussions for the surface energy and water balance, ecosystems, the carbon cycle, and soil nutrient exchange. Despite its importance, the response of soil freeze depth to climate change is largely unknown. This study employs the Stefan solution, and observations from 845 meteorological stations to investigate the response of variations in soil freeze depth to climate change across China. Observations include daily air temperatures, daily soil temperatures at various depths, mean monthly gridded air temperatures, and the Normalized Difference Vegetation Index. Results show that soil freeze depth decreased significantly at a rate of -0.18±0.03 cm/year, resulting in a net decrease of 8.05±1.5 cm over 1967–2012 across China. On the regional scale, soil freeze depth decreases varied between 0.0 and 0.4 cm/year in most parts of China during 1950-2009. Investigating potential climatic and environmental driving factors of soil freeze depth variability, we find that mean annual air temperature and ground surface temperature, air thawing index, ground surface thawing index, and vegetation growth are all negatively associated with soil freeze depth. Changes in snow depth are not correlated with soil freeze depth. Air and ground surface freezing index are positively correlated with soil freeze depth. Comparing these potential driving factors of soil freeze depth, we find that freezing index and vegetation growth are more strongly correlated with soil freeze depth, while snow depth is not significant. We conclude that air temperature increases are responsible for the decrease in seasonal freeze depth. These results are important for understanding the soil freeze/thaw

dynamics and the impacts of soil freeze depth on ecosystem and hydrological process.
**1 Introduction**

Combining multiple land and ocean surface temperature datasets, the global mean air

temperature increased 0.85 °C over 1880-2012 (Stocker et al., 2014). Given that all of the
cryosphere's components are inherently sensitive to air temperature changes on different time scales,
cryospheric changes serve as indicators of climate change. Frozen ground is an important
component of the cryosphere. Permafrost regions underlay approximately 24% of the exposed land
surface of the Northern Hemisphere (Zhang et al., 1999), and seasonally frozen ground (SFG)
regions occupy 57% (Zhang et al., 2003). China has the third-largest frozen ground extent in the
world, with a permafrost area of ~$2.20 \times 10^6$ km$^2$, or approximately 23% of its land area, mainly on
the Tibetan Plateau; regions with SFG occupy about 50% of the land area in China (Zhou et al.,
2000). Under warming climate conditions, frozen ground regions are vulnerable to subsidence,
especially ice-rich permafrost and relatively warm discontinuous permafrost (Morison et al., 2000;
Osterkamp et al., 2000; Stendel and Christensen, 2002). Maximum soil freeze depth of SFG and
active layer depth over permafrost play a significant role in cold environments, and all hydrological,
ecological, biological, and pedological activities occur within this layer (Hinzman et al., 1991; Kane
et al., 1991; Zhao et al., 2004). Simultaneously, soil freeze depth influences the surface and
subsurface hydrologic cycle, promotes soil texture changes, and alters the availability of soil
nutrients for plant growth. The soil freeze/thaw cycle and soil freeze depth variations affect the
decomposition of soil organic matter and greenhouse gas exchanges between the land surface and
the atmosphere (Shiklomanov and Nelson, 2002; Mu et al., 2015; Jafarov and Schaefer, 2016). Thus,
seasonal soil freeze depth variability and climate are closely linked.

Due to global climate warming, significant efforts have been devoted to permafrost research,

such as permafrost variations on the hemispheric-scale, permafrost temperature changes (Wu and
Zhang, 2008; Romanovsky et al., 2010; Guglielmin and Cannone, 2012; Streletskiy et al., 2014; Wu
et al., 2015), permafrost degradation (Jorgenson et al., 2006; Ravanel et al., 2010; Sannel and Kuhry,
2011; Streletskiy et al., 2015; Park et al., 2016), hydrological processes in permafrost regions (Hu
et al., 2009; Wang et al., 2009; Park et al., 2013; Streletskiy et al., 2015; Ford and Frauenfeld, 2016),
feedbacks to climate change (Schuur et al., 2008; Park et al., 2015; Abbott et al., 2016), and other

aspects. The increasing thickness of the active layer has been indicated by many observations in permafrost regions at high latitudes (Brown et al., 2000; Frauenfeld et al., 2004; Zhang et al., 2005; Fyodorov-Davydov et al., 2008; Smith et al., 2010; Wu and Zhang, 2010; Zhao et al., 2010; Callaghan et al., 2011; Li et al., 2012; Liu et al., 2014; Stocker et al., 2014). Less research has focused on SFG areas (Zhang et al., 2003; Frauenfeld et al., 2004; Frauenfeld and Zhang, 2011; Wang et al., 2015), although the near-surface soil freeze/thaw status has been investigated using satellite passive microwave remote sensing (Zhang and Armstrong, 2001; Zhang et al., 2003, 2004; Li et al., 2008; Jin et al., 2015). Peng et al. (2016) analyzed the response of soil freeze/thaw states to climate change across China, based on observational data. While Peng et al. (2016) investigated the area extent changes of different soil freeze/thaw states, here we instead focus on seasonal soil freeze depth. Regional-scale soil freeze depth can be an important indicator of climate change and frozen ground condition in cold regions. Further, SFG is closely related with human activities, because most populated areas are located on SFG.

Shiklomanov (2012) similarly pointed out that SFG has not received much attention despite its vast area extent and importance, mainly due to a lack of long-term observational time series to document changes. Evaluating climatic and environmental changes on SFG requires comprehensives spatial assessments of available soil temperature records (Shiklomanov, 2012). To date, no comprehensive investigation of soil freeze depth in relation to climate change has been conducted in China, despite the prevalence of SFG in this part of the world. Therefore, using long-term observational data, the goals and unique contributions of this study are 1) to estimate the spatial and temporal variations of seasonal soil freeze depth across China; 2) to quantify the potential forcing factors of soil freeze depth including climatic and environmental factors; and 3) to establish how soil freeze depth variability responds to climate change in China.

**2 Data and methods**

**2.1 Data**

Several datasets are used including daily air and ground surface temperature, daily soil temperature at 0-320 cm depth, mean monthly gridded air temperature, and daily snow depth. In addition, we incorporate a 1-km resolution digital elevation model (DEM) and normalized differential vegetation index (NDVI) data. All datasets are described in detail below.

### 2.1.1 Mean daily air and ground surface temperature

Mean daily air temperature and ground surface temperature data are collected from the China Meteorological Administration (CMA) for a total of 839 meteorological stations (Figure 1) available four times daily at 02:00, 08:00, 14:00, and 20:00 (http://cdc.cma.gov.cn/; Wang et al., 2015). These data come already quality controlled, and station observations date back to the 1950s and 1960s. Some stations end during the 1990s, while others are available until 2013. Most stations are located in east central China, with fewer sites in the west and at high elevations, such as on the Qinghai-Tibetan Plateau (Figure 1). These mean daily air and ground surface temperatures are used to estimate temperature changes and to calculate the freeze/thaw index.

### 2.1.2 Soil temperature

Daily soil temperature data are available for 845 sites across China (Figure 1) from the CMA, measured at the depths of 0.00, 0.05, 0.1, 0.15, 0.2, 0.4, 0.5, 0.8, 1.6, and 3.2 m. The temporal record varies for these stations, with some observations dating back to the late 1950s, and some only to the 1970s. Some station records end in the 1990s, while others are available through 2006 (Wang et al., 2015). Soil temperature is used to calculate the soil freeze depth; we combine the potential maximum soil seasonal freeze depth in permafrost regions, and the maximum soil freeze depth in SFG. The number of stations with both daily air temperature and soil temperature observations is 729.

### 2.1.3 Mean monthly gridded air temperature

Mean monthly gridded air temperature (MMGAT) was used to analyze soil freeze depth at the regional scale across China. We obtained the University of Delaware's 1900–2014 terrestrial air temperature gridded monthly time series (http://climate.geog.udel.edu/~climate/), with a $0.5° \times 0.5°$ spatial resolution. This dataset was produced by combining many observational station records across the world, using spatial interpolation and cross-validation procedures (Legates et al., 1990; Willmott et al., 1995; Peterson et al., 1997, 1998). MMGAT during 1950–2010 is used for assessing its correspondence with seasonal freeze depth across China.

### 2.1.4 Digital elevation model (DEM)

Considering the complex terrain across China and the impacts of elevation on air temperature, we also used the global 30 arc-second elevation dataset (GTOPO30;

https://lta.cr.usgs.gov/GTOPO30) as the digital elevation model (DEM) for this study to further improve the MMGAT resolution. GTOPO30 was derived from several raster and vector sources of topographic information. Across China, the elevation ranges from −152 to 8752 m (Figure 1). Based on this DEM, we spatially interpolate the MMGAT data to the DEM's 30 arc-second (1-km) resolution.

**2.1.5 Snow depth**

We obtained daily mean snow depth data for 672 sites across China (Che et al., 2008). The period of record at these locations varies, with some stations dating back to the late 1950s and some only to the 1970s. Some station records end around the 1990s while others are available through 2005. The snow depth was used to assess its influence on soil freeze depth. We calculate the annual maximum snow depth (SND) from the daily data for 1 July–30 June, and match those snow depth stations with the soil temperature stations. If there are missing data in the spring, autumn, and winter season of one station, this station data will not be used.

**2.1.6 Normalized differential vegetation index (NDVI)**

The NDVI dataset used in this study is produced by the Global Inventory Modeling and Mapping Studies (GIMMS) team, available for 1982–2006. It is derived from NOAA AVHRR data, available at 15-day temporal resolution and an 8-km spatial resolution (Tourre et al., 2008). These data were used to assess the influence of vegetation on soil freeze depth. We extracted the NDVI values corresponding to the stations' latitude and longitude coordinates.

**2.2 Methods**

Missing data often present a potential problem for analyzing and averaging time series. Therefore, if fewer than five days were missing in a given month, filling in missing daily air temperatures was based on highly correlated neighboring sites using linear regression. Missing daily mean ground surface temperatures were estimated through linear regression with the daily mean air temperature at the same station. Based on the daily air temperature, we also calculate the mean monthly air temperature and mean annual air temperature (MAAT). The interpolated results are strongly correlated with observations, as indicated by regression coefficients larger than 0.95.

To improve the original 0.5° × 0.5° MMGAT data to a 1-km resolution, spatial interpolation was used in conjunction with monthly lapse rates and the 1-km resolution DEM (e.g., Willmott and

Matsuura, 1995; Gruber et al., 2012). The data processing steps are to (1) calculate the average
monthly atmospheric lapse rate based on all available meteorological stations across China and their
elevations; (2) bring each average monthly gridded air temperature value to a reference level
(elevation of 0 m) using the average monthly lapse rate; (3) apply a Kriging interpolation to the
reference-level adjusted MMGAT; and (4) bring the gridded reference-level air temperature back to
the DEM-gridded height. Based on more than 800 sites, we evaluated the interpolated MMGAT
against the observational monthly air temperatures, and find that the regression coefficient is almost
1.0 with a minimum of 0.98 in April.
The freezing/thawing index can also be an important indicator to assess the variations in
frozen ground (Zhang et al., 1997; Nelson, 2003; Frauenfeld et al., 2007). There are two primary
types of freezing/thawing indices: the surface freezing/thawing index, calculated from ground
surface temperatures, and the air freezing/thawing index, computed from air temperatures. To
calculate the freezing index, we sum all temperatures below 0 °C during the freezing periods
(equation 1), and similarly calculate the thawing index by summing the above-0 °C temperatures
during the warm season (equation 2; Wu et al., 2011; Luo et al., 2014). We define the freezing period
to be July–June, to sum the freezing index over a continuous cold season (equation 1). The warming
period is defined as the calendar year (Wu et al., 2011; Peng et al., 2013, 2016) (equation 1, 2). Thus,
freezing/thawing index at the point scale was calculated based on the daily mean air temperatures
and ground surface temperatures. For the regional-scale air freezing index, we use the adjusted 1-
km gridded terrestrial air temperature data.
$\text{FI} = \sum_{i=1}^{N_F} |T_i|, T_i < 0\,℃ \quad (1)$
$\text{TI} = \sum_{i=1}^{N_T} T_i, T_i > 0\,℃ \quad (2)$
where $N_F$ is the number of days with temperature below 0 °C; i = 1, 2 … $N_F$; $N_T$ is the number of
days with temperatures above 0 °C; and $T_i$ represents the temperature on a specific day.
Various methods are available to calculate the soil freeze depth. For example, it can be
estimated directly from soil temperature, from physical and statistical models, and based on the
Stefan solution. In this study, we use the Stefan solution to estimate soil freeze depth, which is
determined using equation 3:
$$SFD = \sqrt{2K_f(\frac{n_f FI_a}{P_b wL})} \quad (3)$$
where SFD is soil freeze depth (m), $K_f$ is the thermal conductivity of the frozen soil (W/m • °C), $n_f$
is the n-factor for the freezing season and corresponds to the ratio between the surface freezing
index and the air freezing index (Peng et al., 2016), $FI_a$ is the annual air freezing index (°C•d), $P_b$ is
the soil bulk density (kg/m$^3$), w the soil water content by weight, and L the latent heat of fusion
(J/kg) (Zhang et al., 2005). In equation 3, many site-specific factors are required to estimate SFD,
which are generally not available, particularly at the regional scale. However, based on the SFD and
annual freezing index at each observational site, we can quantify the relationship between these two
parameters (Figure 2). We find a strong and statistically significant correlation of R=0.87. Thus, the
relationship between SFD and the annual freezing index can be simplified (Harlan and Nixon, 1978)
as:
$$SFD = E\sqrt{FI_a} \quad (4)$$
where E is defined (Nelson and Outcalt, 1987) as:
$$E = \sqrt{\frac{2K_f n_f}{P_b wL}} \quad (5)$$
To estimate the SFD at the regional scale across China, we first calculate SFD for every
observational station by interpolating the depth of the 0°C isotherm throughout the 0.0–3.2 m soil
profile using the daily mean soil temperature (Frauenfeld et al., 2004). Next, we estimate the $FI_a$
based on the calculations in Frauenfeld et al. (2007). To estimate the E value for all stations, we use
the SFD, $FI_a$, and equations 2 and 3. Then, we interpolate the E value to the regional scale at 1-km
resolution using kriging in ArcGIS. The SFD is estimated across China based on equation 4, the 1-
km E value, and $FI_a$. We can then estimate the regional-scale SFD for each year from 1950 to 2009
across China, and obtain the mean decadal SFD. Finally, we estimate the SFD trend at the regional
scale across China based on regression analysis.
From the 1-km scale E factor values, we can extract every site's E factor based on the sites'
latitude and longitude. Then, the air freezing index from the sites is used to calculate the annual soil
freeze depth at every site by equation 4. To evaluate the result, we compare the observational SFD

calculated from the soil temperatures, and the simulated SFD derived from the Stefan method (equation 4) in figure 3. The result demonstrates that the mean absolute error and root-mean-square error are 0.08 m and 0.14 m, respectively, indicating that there is a good agreement between simulated and observational SFD.

A number of climatic and environmental variables including MAAT, mean annual ground surface temperature (MAGST), freezing index, thawing index, SND, and NDVI are selected to investigate the potential drivers of the observed long-term SFD changes across China. We use Pearson correlations to analyze the association between these variables and SFD, and employ a 95%-significance level to assess the statistical significance for all analyses.

## 3 Results

### 3.1 Soil freeze depth

Figure 4 shows the spatial variability and trends of SFD at every location. The highest SFD was mainly located in northeastern and northwestern China, and the Tibetan Plateau. In contrast, the lowest SFD was found in the south of China. Locations with SFD greater than 0.4 m are found north of the Yellow River. In the northwest of China, locations with SFD less than 0.8 m are found in the Taklimakan desert, and some sites with SFD greater than 2.0 m are located in the Altai, Tianshan, and Pamir Mountains.

On the Tibetan Plateau, most sites have a SFD greater than 2.4 m. There is an increase in SFD with increasing latitude and elevation. The significant SFD changes are between -0.4 and less than 0 cm/year. The sites with the strongest decreasing trends of -1.2 cm/year are on Tibetan Plateau and -1.0 cm/year in the north of China.

Figure 5 shows the standard deviation of SFD at each site across China. It varies from 0.00– 0.27 m. The standard deviation of SFD is generally less than 0.03 m south of 35°N, except on the Tibetan Plateau. In northeastern China, the standard deviation changes between 0.06 m and 0.15 m. In the northwest, it is generally 0.06–0.12 m. On the Tibetan Plateau, the standard deviation varies from less than 0.09 m, but can be greater than 0.18 m at some sites.

Based on the sites' E factors and $FI_a$, we calculate SFD time series anomalies from 1951 to 2012 (Figure 6). Although a composite time series of all available stations data can be calculated during 1951–2012, few of 839 stations actually contribute to the mean values before the 1960s

(Figure 6). There are fewer than 200 stations in the early years, which therefore does not represent
the SFD across China as a whole. Beginning in 1967, more than 800 stations contribute to each
year's mean, therefore long-term SFD trends will only be evaluated from then on. There is a
statistically significant change in SFD anomalies of -0.18±0.03 cm/year, corresponding to a net
decrease of 8.05±1.5 cm. In addition to the overall long-term decrease, there are also some patterns
of inter-decadal variability during 1967-2012, including slight positive changes in some periods.
SFD exhibited both increases and decreases until 1975, followed by a sharp decrease until 1990.
However, SFD has remained constant or may perhaps be increasing slightly during 1990-2012.
Therefore, the overall SFD change during 1967–2012 was largely controlled by the decrease during
1975–1990. Similar SFD changes, attributable to variability in the North Atlantic Oscillation, were
found in high-latitude Eurasia (Frauenfeld and Zhang, 2011).
**3.2 Spatial and temporal variability of SFD in China**
Based on the 1-km resolution E factor and 1-km $FI_a$ calculated from MMGAT, we estimate
SFD across China from 1950 to 2009 by the Stefan method. Figure 7 shows the spatial **pattern of**
**multi-year** mean SFD. SFD increases with latitude and elevation, with SFD greater than 1.5 m in
northeastern China, the Mongolia Plateau, Tibetan Plateau, and north of the Xinjiang region. In the
east of China, the SFD ranges from 0.0 m to more than 4.0 m, and increases with latitude. In the
Yellow River region, the elevation decreases from west to east, while the SFD varies from greater
than 2.5 m to less than 0.5 m. The SFD in the Taklimakan desert is lower than in the surrounding
area. **Figure 8 demonstrates the spatial variability of SFD anomaly for the decades of the 1950s,**
**1960s, 1970s, 1980s, 1990s, and 2000s, with respect to the 1950–2009 mean across China.**
**Results show that the spatial changes of SFD anomaly ranges from larger than 0.15 m to less**
**than -0.1 m. From 1950s to 2000s, the SFD anomaly changes from positive to negative, which**
**means that SFD has a decrease trend during this period. In addition, the smaller variability**
**of SFD anomaly occurs in south of Yellow River and desert regions in Xinjiang.**
Figure 9 represents the SFD trend across China from 1950 to 2009. The gray region
represents areas where the SFD trends are not statistically significant, however, they are statistically
significant in all other regions. In general, the SFD decreased significantly over northern China,
except in two small areas. The SFD trend ranges between 0.0 and -0.4 cm/year in most areas. SFD
trends less than -0.4 cm/year are found in some areas, such as the Tibetan Plateau, and the Pamirs.
In the two small areas of increasing SFD, we further investigated the MAAT trend during 1950-
2010 based on the MMGAT dataset. There is similarly a statistically significant decrease of MAAT
in these same areas during this period. Therefore, air temperature is possibly one of the important
factors that influence SFD in these areas. More detailed discussion is provided in sections 3.3 and

4.1.

Overall, the spatial variability indicates that SFD changes with latitude and elevation at the
regional scale across China. As is expected from climate warming, a statistically significant
decreasing trend in SFD is evident across China from 1950 to 2009.
**3.3 Potential forcing variables**
To explore the possible variables leading to the documented changes in SFD, we analyze
potentially important factors for soil freeze dynamics: latitude, altitude, MAAT, MAGST, freezing
index including $FI_a$ and the ground surface freezing index ($FI_s$), thawing index including the air ($TI_a$)
and ground surface thawing index ($TI_s$), SND, and NDVI.
**To explore the spatial variability of SFD, we classify the meteorological stations as**
**either eastern or western based on 110°E longitude. Figure 10 represents the correlations**
**between SFD and latitude and altitude in the eastern and western parts. In the east, we find**
**an exponential relationship between SFD and latitude, and a linear relationship with altitude,**
**with both being statistically significant. The SFD values range from 0.0 m to less than 3.5 m,**
**varying with latitude more so than with altitude. Thus, SFD is mainly affected by latitude in**
**eastern China. In the west, SFD is near 0.0 m where altitude is higher than 1000 m. Similarly,**
**SFD is related statistically significantly with altitude and latitude, but altitude is the main**
**factor affecting SFD in the west.**
Temperature—including MAGST and MAAT—at the 839 station locations exhibits a
statistically significant increase over the 1951–2013 period of 0.019 and 0.013 °C/year, or
approximately 1.2 °C and 0.78 °C over the 63 years, respectively (Figure 11 a, and b). MAGST and
MAAT are statistically significantly correlated with SFD at R=-0.56 and R=-0.66, which means that
31% and 44%, respectively, of the variability in SFD can be accounted for by these temperature
measures. Further, the negative correlation demonstrates that increasing temperatures result in SFD
decreases at the 839 stations.

Soil freeze usually begins in autumn or winter, with temperatures less than 0 °C reaching

their maximum freeze depth toward the end of winter season or spring. Therefore, maximum annual
SFD occurs during the cold seasons. Freezing index is an important indicator for accumulated cold
season temperatures (Frauenfeld and Zhang, 2011). From 1951 to 2013, $FI_s$ and $FI_a$ underwent a
statistically significant decrease of 3.0 and 1.62 °C-days/year, respectively (Figure 11 c, and d),
indicating warming, which reduces the cold season's magnitude and/or duration. The correlation
between $FI_s$, $FI_a$, and SFD was a statistically significant 0.68 and 0.87, indicating that the FI accounts
for 46% and 76% of SFD variability.

The thawing index is used to assess the accumulated positive degree-days during the warm

season (Frauenfeld and Zhang, 2011). There are no obvious TI changes at the station locations until
approximately 1985. TI increases during 1985-2008, followed by a decrease until 2013. From 1951
to 2013, $TI_s$ and $TI_a$ show statistically significant increases at a magnitude of 3.73 and 2.77 °C-
days/year, respectively (Figure 11 e and f). The correlation coefficient between $TI_s$, $TI_a$, and SFD is
-0.53 and -0.57, respectively, indicating a weak negative association, such that warm summer
conditions correspond to a shallower SFD the following cold season.

Figure 12 shows the correlation between SFD and SND. However, the weak negative

correlation between SFD and SND of R=-0.13 is not statistically significant, indicating that there is
no relationship. This result is also consistent with the findings of Jafarov and Schaefer (2016).

As suggested by Shiklomanov (2012), environmental factors likely also affect SFD. The

surface can be affected directly by climate forcing, while the subsurface effects are more complex.
The subsurface soil only indirectly receives a climatic signal, which is furthermore altered by site-
specific soil processes (e.g., thermal conductivity and analogous soil properties). Vegetation is a
likely environmental factor that influences the soil freeze depth (Shiklomanov, 2012). Thus, we
investigate vegetation using NDVI (Peng et al., 2013) and find it is significantly correlated with
SFD at -0.80, suggesting that 64% of the variability in SFD can be accounted for by NDVI. The
statistically significant negative correlation demonstrates that when NDVI increases (greening), this
corresponds to a decrease in SFD (Figure 13).
**4 Discussion**

Soil freeze/thaw depth changes involve a series of interactions, such as energy exchanges,

soil moisture exchanges, and gas exchanges between the atmospheric and terrestrial system.
Therefore, variations of soil freeze/thaw most likely have an important effect on geomorphic,
hydrological, and biological processes. Similarly, soil freeze/thaw depth changes also have
destabilizing effects on engineering structures, such as on improperly constructed infrastructure
(Smith and Burgess, 1999; Stendel and Christensen, 2002). The release of additional greenhouse
gases to the atmosphere also occurs (Michaelson et al., 1996; Mu et al., 2015). In this paper, we use
the Stefan method to calculate SFD, analyze the spatial SFD variability and trends, and quantify the
potential driving factors affecting SFD.
**4.1 Climatic and environmental factors**

SFD variability is susceptible to climate warming and environmental change, and is

affected by variables including air temperature, ground surface temperature, freezing/thawing index,
and vegetation. Many examples of permafrost degradation have been reported, such as deeper the
active layer thickness, reduced freeze time duration, and shifts in the timing of thawing and freezing
in seasonally frozen ground regions (Henry, 2008; Callaghan et al., 2011; Stocker et al., 2014; Wang
et al., 2015). Negative correlations are found here between SFD and temperature (including MAAT
and MAGST), because of solar radiation heating the ground, energy transfer into the soil, ultimately
increasing the soil temperature. Thus, increasing temperature is found to be the main factor
influencing SFD variability in China, as in previous work focusing only on the Tibetan Plateau
(Zhao et al., 2004).

The freezing/thawing indices represent the accumulated negative and positive degree-days

in the cold and warm seasons, respectively (Wu et al., 2011). The positive and negative correlation
between SFD and FI and TI were statistically significant, consistent with previous results in other
regions (Frauenfeld and Zhang, 2011). Due to the maximum soil freeze depth occurring in the cold
season and SFD being affected by temperature, the positive correlation between SFD and FI is
reasonable. Although TI is the accumulated temperature in the warm season, it takes some time to
transfer the energy into the deeper ground. The energy flux into the soil reduces with increasing soil
depth. Therefore, if all the conditions are the same, a larger TI can precondition the ground by
increasing the energy in the deeper soil, which can subsequently delay soil freezing. Thus TI is a
potential indicator of SFD, indirectly affecting soil temperature (Frauenfeld and Zhang, 2011).
Snow depth can have an effect on soil temperature, which would affect the active layer
thickness and seasonal SFD variability. Numerical modeling studies have shown that snow depth
does impact SFD (Zhang and Stamnes, 1998; Ling and Zhang, 2003; Park et al., 2015). Park et al.
(2015) indicated that both increasing SND and snow structure (e.g., snow density) changes were
favorable to soil warming, resulting in active layer thickness decreasing in northern regions as
previously found by Frauenfeld et al. (2004). Snow cover insulates the ground during the cold
season (Zhang, 2005). Interestingly, in our study we did not find a relationship between SND and
SFD. This could be due to the spatial heterogeneity of snow across China. According previous
research, snow depth, snow water equivalent, and snow densities are smallest on the Tibetan Plateau
compared to other parts of China (Ma et al., 2012). Compared with other regions, multi-year average
snow depth in general is low in China, especially on the Tibetan Plateau and the east-central
mountain regions of China (Zhong et al., 2014), and may therefore have only limited insulating
effects. This could lead to the lack of a relationship between SFD and SND across China and
motivates future investigation.
A negative correlation between SFD and vegetation, as quantified by NDVI, is found.
Vegetation change has a significant influence on the climate system mostly through changes to the
surface radiative energy budget, which can be affected the SFD. Based on previous research,
vegetation varies in different land cover types and responds to climate change via different physical
mechanisms (Snyder et al., 2004), e.g., changes in the surface albedo (e.g., bare ground versus
vegetation cover), vegetation transpiration, and shading effects (Kelley et al., 2004; Snyder et al.,
2004; Swann et al., 2010; Chang et al., 2012; Zhang et al., 2012). In the cold season, less/decreased
vegetation will be more easily snow covered, thus increasing the albedo considerably. Increasing
albedo results in less net radiation at the land surface, as more incoming solar radiation is reflected
from the surface. Then, the surface air temperature will decrease considerably due to less energy
absorbed at the surface. For the colder land surface, the sensible heat flux is reduced. Further, the
vegetation decrease results in reducing evapotranspiration, which decreases the latent heat flux
(Snyder et al., 2004). Compared to increased vegetation cover, less vegetation causes a large annual-
average increase in the surface albedo with the largest changes in the winter and spring seasons,
which reduces the amount of net radiation at the surface, making the surface colder and resulting in
SFD increases. Conversely, vegetation increases could lead to decreasing SFD. The vegetation's
effect on transpiration is primarily important in summer, while SFD primary occurs in winter and
spring (Snyder et al., 2004).

The **inverse relationship** between NDVI and SFD **is in agreement with results** from many

previous studies that **similarly found** a vegetation increase, or a greening trend, in different regions
during **recent** decades (Peng et al., 2011; Piao et al., 2011; Zhang et al., 2013; Zhu et al., 2016).
Because climate change controls the spatial distribution of vegetation, most studies **report**
vegetation **increases** as impacted by temperature and precipitation **increases** (Bao et al., 2015;
Huang et al., 2016). Similarly, figure 9 shows that rising temperature results in a SFD decrease. The
negative relationship between SFD and NDVI indicates the effect of vegetation on SFD, and also
their **inverse** relationship.

SFD is affected by many factors, including the climatic and environmental variables

considered in this study. However, SFD changes in different regions are also potentially influenced
by many other local environmental variables or large-scale teleconnections (**Frauenfeld and Zhang,**
**2011**). Thus, it remains difficult to fully account for the spatial variations of SFD at the regional
scale.
**4.2 Soil Freeze Depth in Different Climate Zones**

Our results indicate significant changes of SFD across China. To address the spatial pattern

of SDF changes, we divide the study area into five different zones, including tropical monsoon
(TPM), subtropical monsoon (SM), temperate monsoon (TM), temperate continental (TC), and
Qinghai-Tibetan Alpine (QTA) climate zones, which are categorized by temperature, precipitation,
and other parameters. Results indicate that the 30-year (1971-2000) average SFD in the SM, TM,
TC, and QTA climate zones are 2.8±0.5 cm, 113.6±7.6 cm, 132.5±7.8 cm, and 165.8±6.7 cm,
respectively. Similar changes of SFD are found across the TM (-0.27±0.005 cm/year), TC (-
0.26±0.005 cm/year), and QTA (-0.22±0.004 cm/year) zones during 1950-2009, while there are no
significant changes in the SM region. This is likely due to the higher temperatures in SM climate
zone (Fig. 12). Although this study investigates a number of environmental and climatic driving
variables of SFD, the degree to which other potential factors (e.g., soil texture, soil moisture, albedo,
cloud cover, teleconnections) could also influence SFD remains unknown due to a lack of reliable
data.

**5 Summary and Conclusions**

In this study, we conducted a comprehensive regional-scale investigation of SFD across
China. A significant climate indicator, SFD is influenced by many variables including climatic and
environmental factors. These factors are often integrated to affect SFD (Lachenbruch and Marshall,
1986; Brown et al., 2000; Frauenfeld et al., 2004). Our results can be summarized as follows:
The spatial distribution of SFD variability is influenced by latitude and elevation across
China. High latitude and altitude sites are characterized by large SFD. In contrast, smaller SFD
values are generally observed for lower latitude and lower elevation regions.
Of the total 839 sites, we find that the SFD decreased significantly, at -0.18±0.03 cm/year
from 1967 to 2012, equal to a net change of 8.05±1.5 cm. The long-term decrease also exhibits inter-
decadal variability, including some positive changes in some periods and no change since 1990.
On the regional scale, the 1950–2009 spatial variation of SFD ranges between 0.0 and 4.5 m
across China, with most areas exhibiting significant decreases between less than 0.0 and -
0.4 cm/year. Different climatic and environmental factors were explored as potential driving
variables of SFD. A negative relationship is evident between SFD and MAAT, MAGST, $TI_a$, and
$TI_s$, with statistically significant correlations of -0.66, -0.56, -0.57, and -0.56, respectively. The
climatic factors $FI_s$ and $FI_a$ were correlated positively with SFD, at 0.87 and 0.68, respectively.
There is no correlation between SFD and SND. The environmental factor vegetation (NDVI) is
negatively correlated with SFD, indicating that 64% of the changes in SFD can be accounted for by
vegetation. Of the potential drivers of SFD explored here, FI and NDVI are most strongly correlated
with SFD, while **SND did not show a significant association**.

**Acknowledgments**: This study was funded by the National Natural Science Foundation of China
(grant No. 91325202, 41601063, 41671516), the National Key Scientific Research Program of
China (grant No. 2013CBA01802), and the Fundamental Research Funds for the Central
Universities (lzujbky-2015-217). We acknowledge computing resources and time at the
Supercomputing Center of Cold and Arid Region Environment and Engineering Research Institute
of Chinese Academy of Sciences.

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

**Figure captions:**
**Figure 1.** The observational station distribution across China, including the 839 stations with air
and ground surface temperatures (green symbols), 845 soil temperature stations (red symbols), and
elevation. The blue solid lines represent the main rivers.

**Figure 2.** Linear least squares regression between soil freeze depth and annual freezing index based on observational sites. The black solid line represents the linear regression.

**Figure 3.** Comparison of the simulated and observed SFD for all stations. The black solid line is the 1:1 line, while the gray dashed line is regression fit between the simulated and observed values.

**Figure 4.** Spatial distribution and variability of SFD at the observing stations. (a) Multi-year mean SFD at each site; (b) the number of sites contributing to the SFD mean; (c) the magnitude of SFD change at each site; (d) the number of sites with SFD change observations.

**Figure 5**. The standard deviation of SFD at each site across China.

**Figure 6.** 1951–2012 SFD anomalies with respect to the 1971–2000 mean (red solid line) based on up to 839 stations across China as depicted in figure 1. Included also is the 1 standard deviation range (gray shading), the linear trend from 1967 to 2012 (blue dashed line), and the 7-year smoothing (green line). The inset shows the number of stations contributing to the time series.

**Figure 7.** Spatial variability of SFD in the decades of the 1950s, 1960s, 1970s, 1980s, 1990s, and 2000s across China.

**Figure 8.** Spatial variability of SFD anomaly for the decades of the 1950s, 1960s, 1970s, 1980s, 1990s, and 2000s, with respect to the 1950–2009 mean across China.

**Figure 9.** SFD trends across China from 1950 to 2009. The grey regions indicate non-significant SFD changes, while trends in all other regions are statistically significant.

**Figure 10.** The relationship between SFD, latitude, and elevation in the east and west of China.

**Figure 11.** SFD time series and trend (black) and the potential forcing variables: (a) mean annual ground surface temperature (red), (b) mean annual air temperature (green), (c) surface freeze index (cyan), (d) air freezing index (magenta), (e) surface thawing index (yellow), (f) air thawing index (orange). All variables are standardized to range from 0–1. R is the correlation coefficient, and all are statistically significant.

**Figure 12.** Correlation between SFD and SND. The variables are standardized to range from 0–1.

**Figure 13.** Correlation between SFD and mean annual NDVI.

**Figure 14.** Time-series of SFD changes in different climate zones: (a) subtropical monsoon, (b) temperate monsoon, (c) temperate continent, (d) Qinghai-Tibetan Plateau Alpine, and (e) tropical. The insets are the SFD changes in the four climate zones; the bold black line is SFD, and bold red line is the trend.

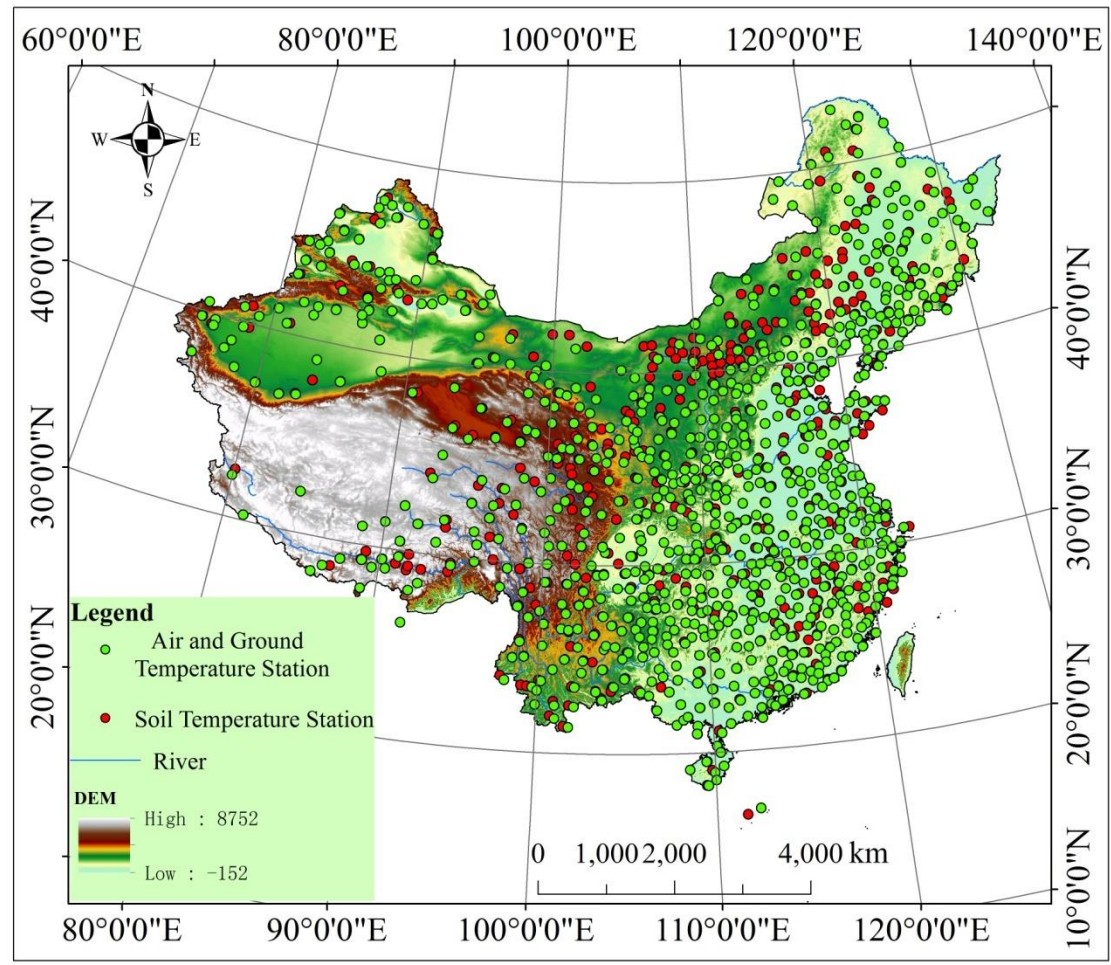

**Figure 1.** The observational station distribution across China, including the 839 stations with air
and ground surface temperatures (green symbols), 845 soil temperature stations (red symbols), and
elevation. The blue solid lines represent the main rivers.

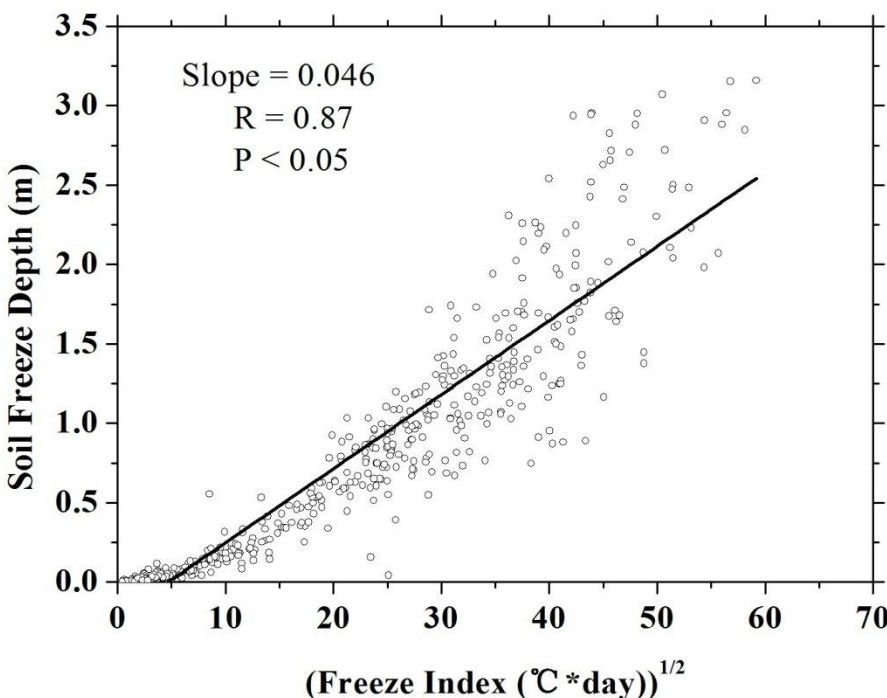

**Figure 2.** Linear least squares regression between soil freeze depth and annual freezing index based
on observational sites. The black solid line represents the linear regression.

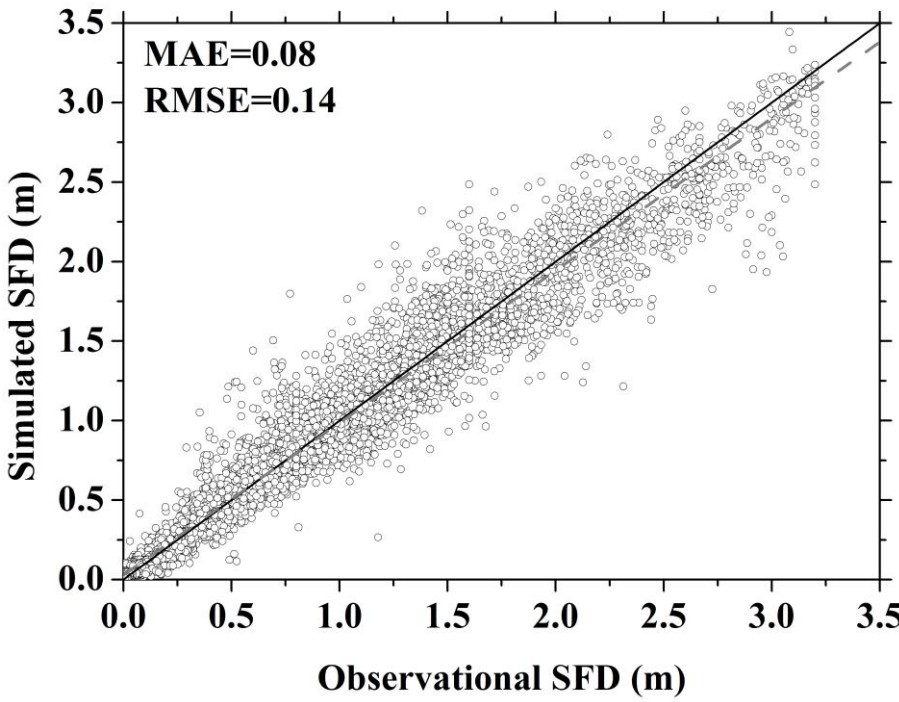

**Figure 3.** Comparison of the simulated and observed SFD for all stations. The black solid line is
the 1:1 line, while the gray dashed line is regression fit between the simulated and observed
values.

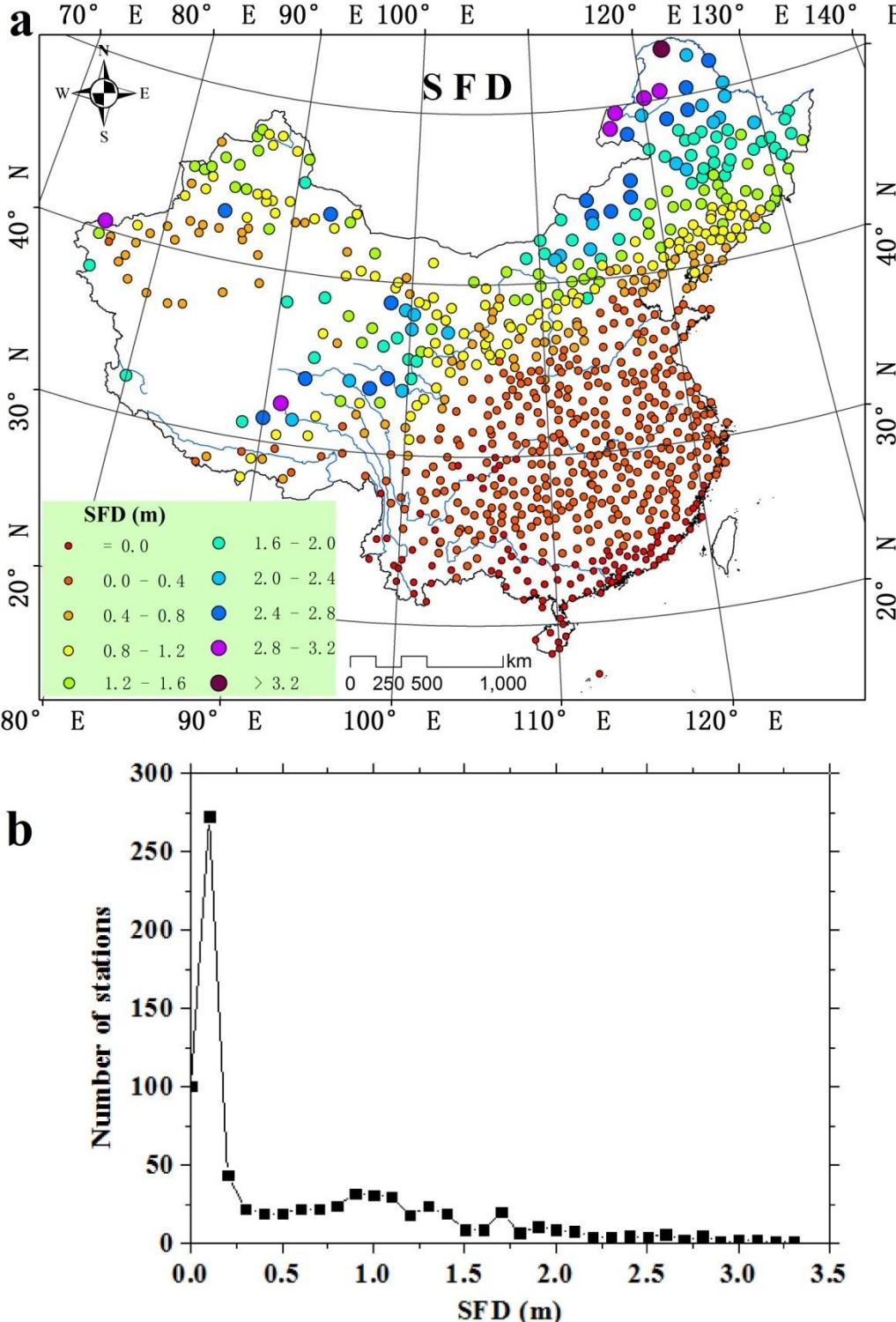


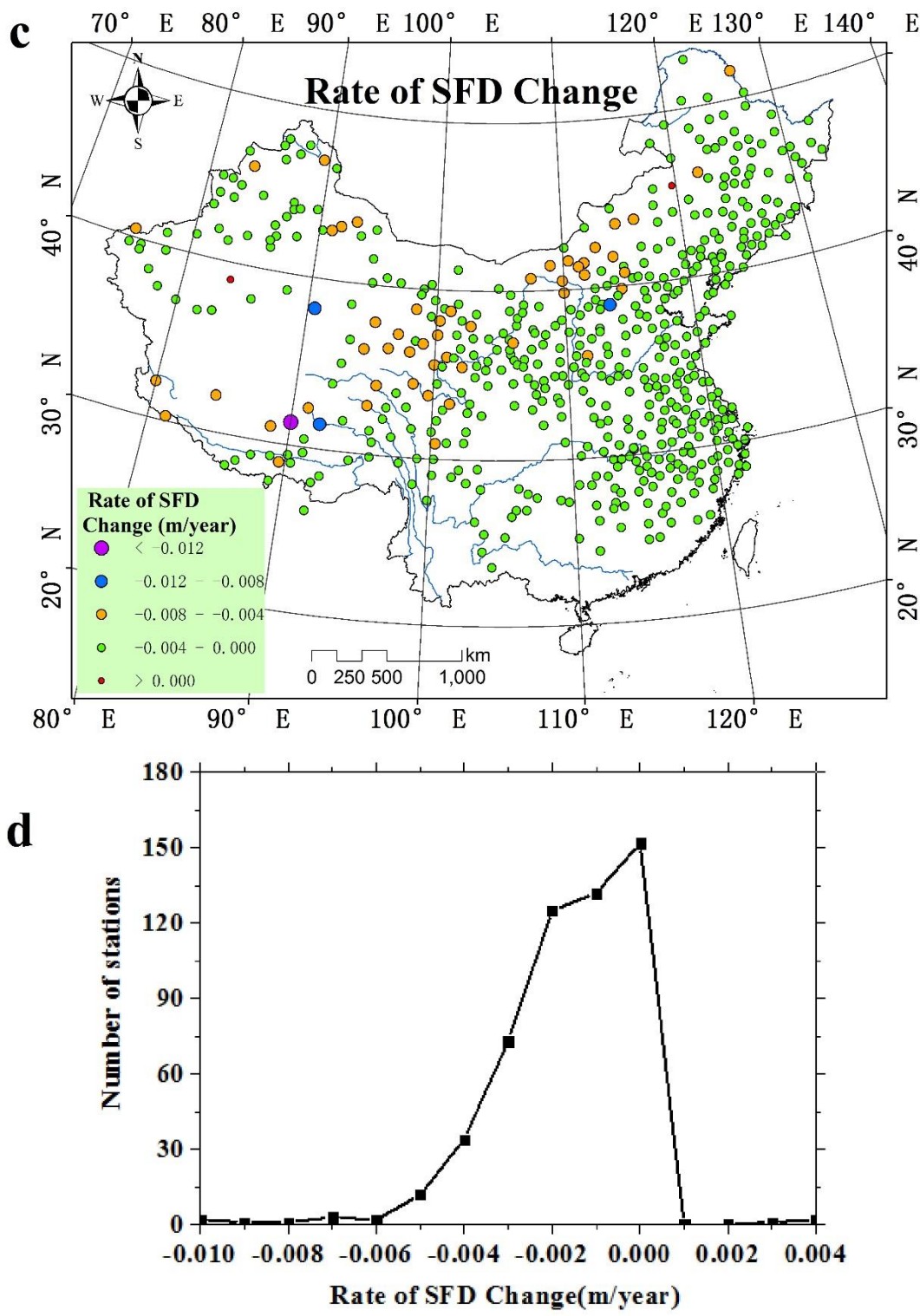

**Figure 4.** Spatial distribution and variability of SFD at the observing stations. (a) Multi-year mean
SFD at each site; (b) the number of sites contributing to the SFD mean; (c) the magnitude of SFD
change at each site; (d) the number of sites with SFD change observations.


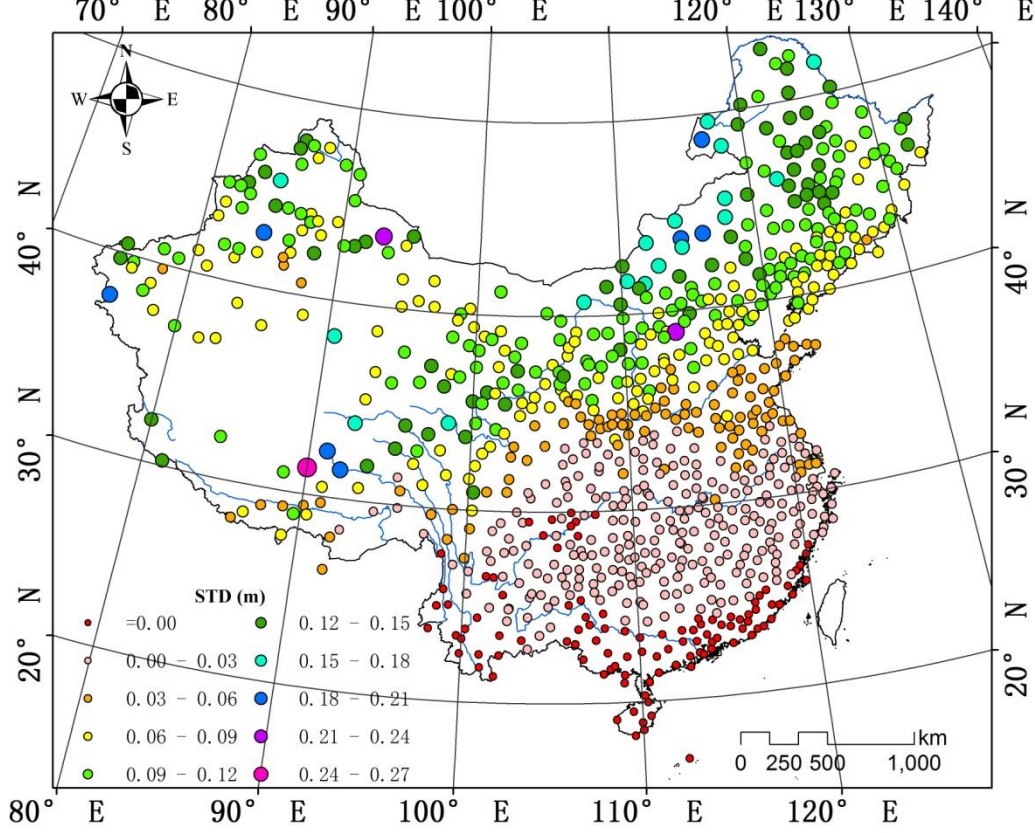

**Figure 5**. The standard deviation of SFD at each site across China.

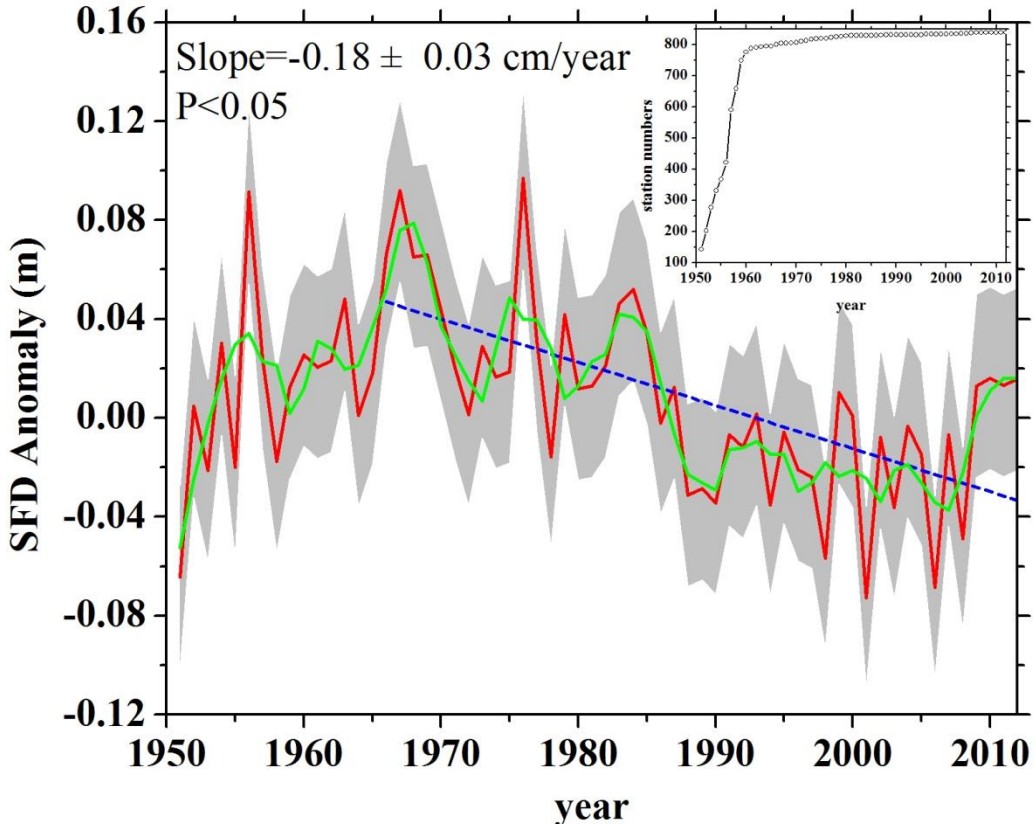

**Figure 6.** 1951–2012 SFD anomalies with respect to the 1971–2000 mean (red solid line) based
on up to 839 stations across China as depicted in figure 1. Included also is the 1 standard deviation
range (gray shading), the linear trend from 1967 to 2012 (blue dashed line), and the 7-year
smoothing (green line). The inset shows the number of stations contributing to the time series.

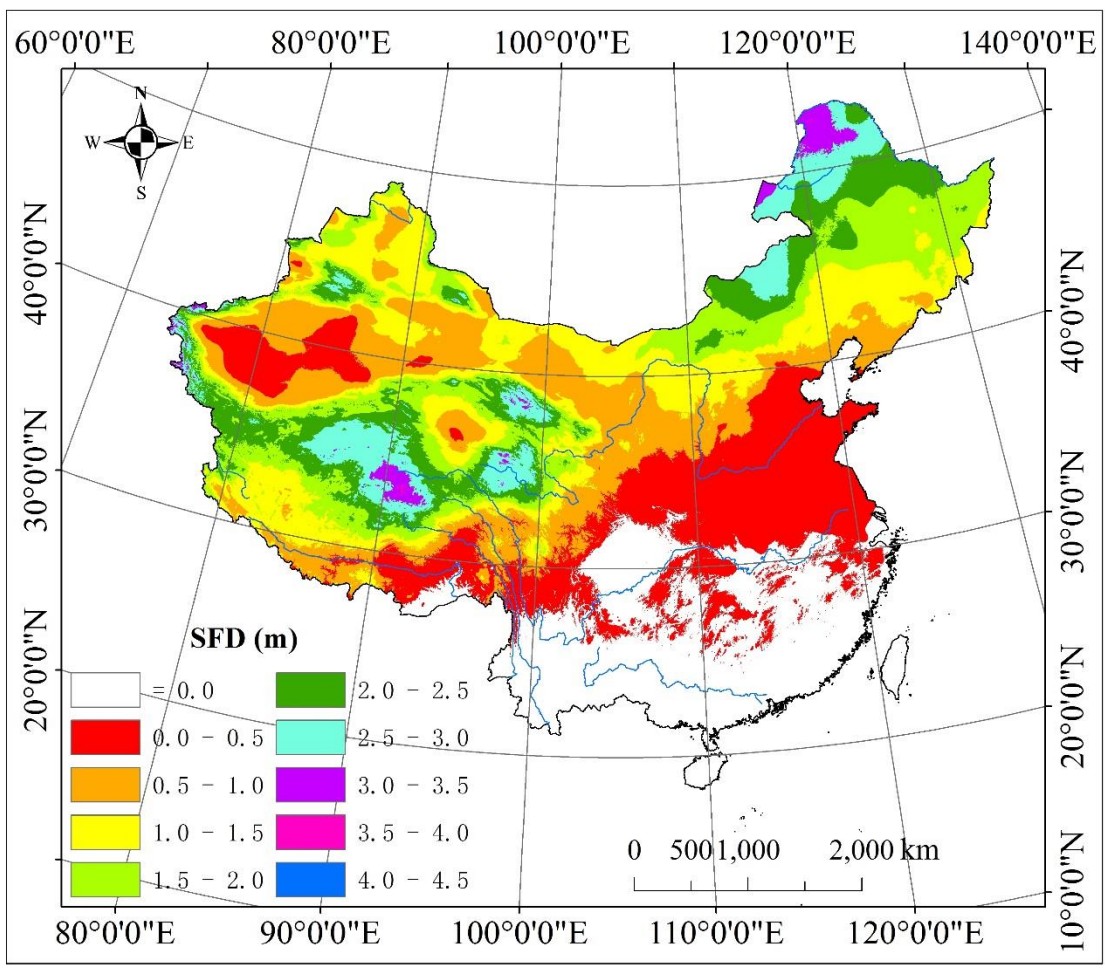

**Figure 7.** Spatial pattern of multi-year mean SFD during 1950-2009 across China.

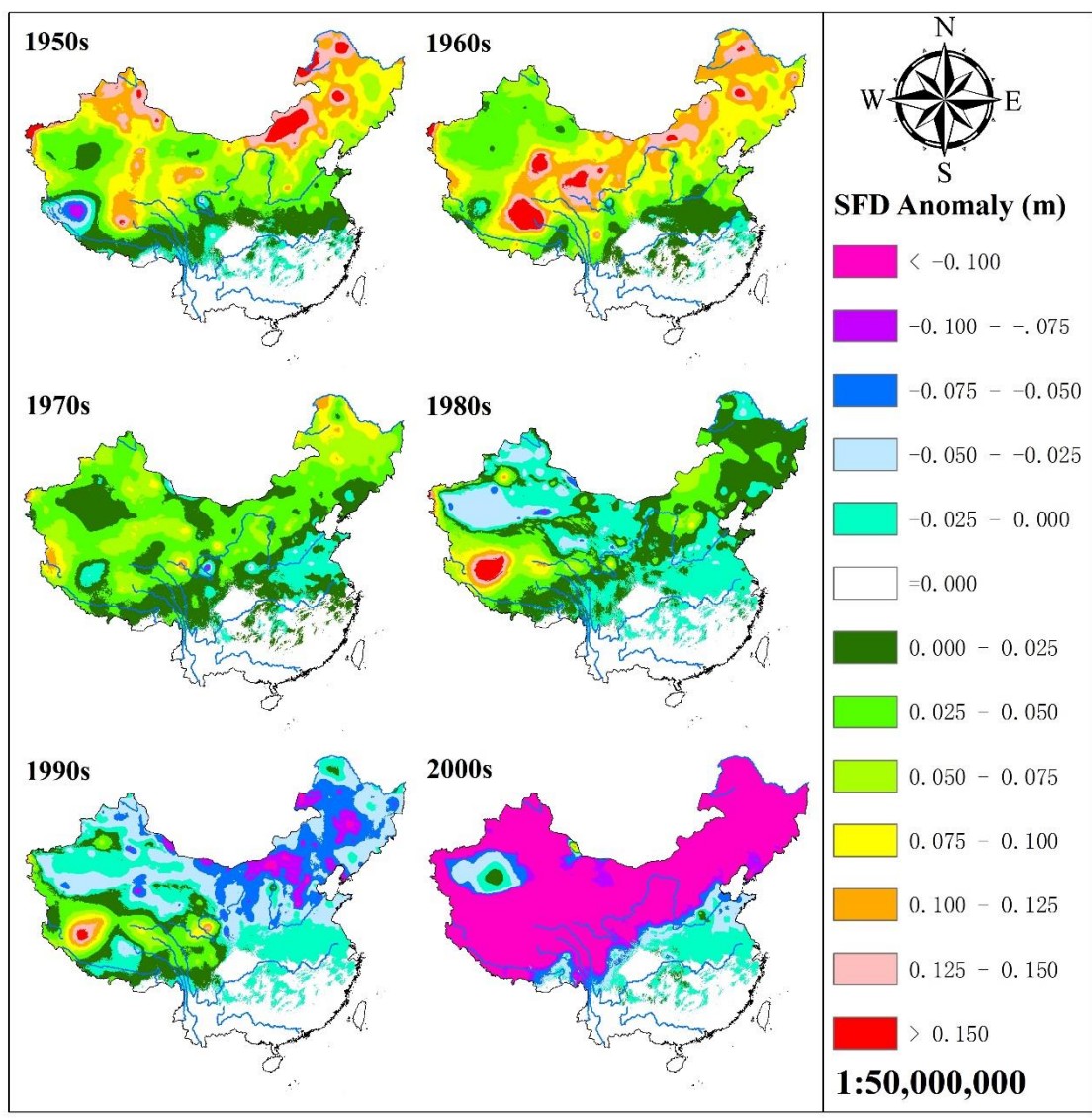

**Figure 8.** Spatial variability of SFD anomaly for the decades of the 1950s, 1960s, 1970s, 1980s,
1990s, and 2000s, with respect to the 1950–2009 mean across China.

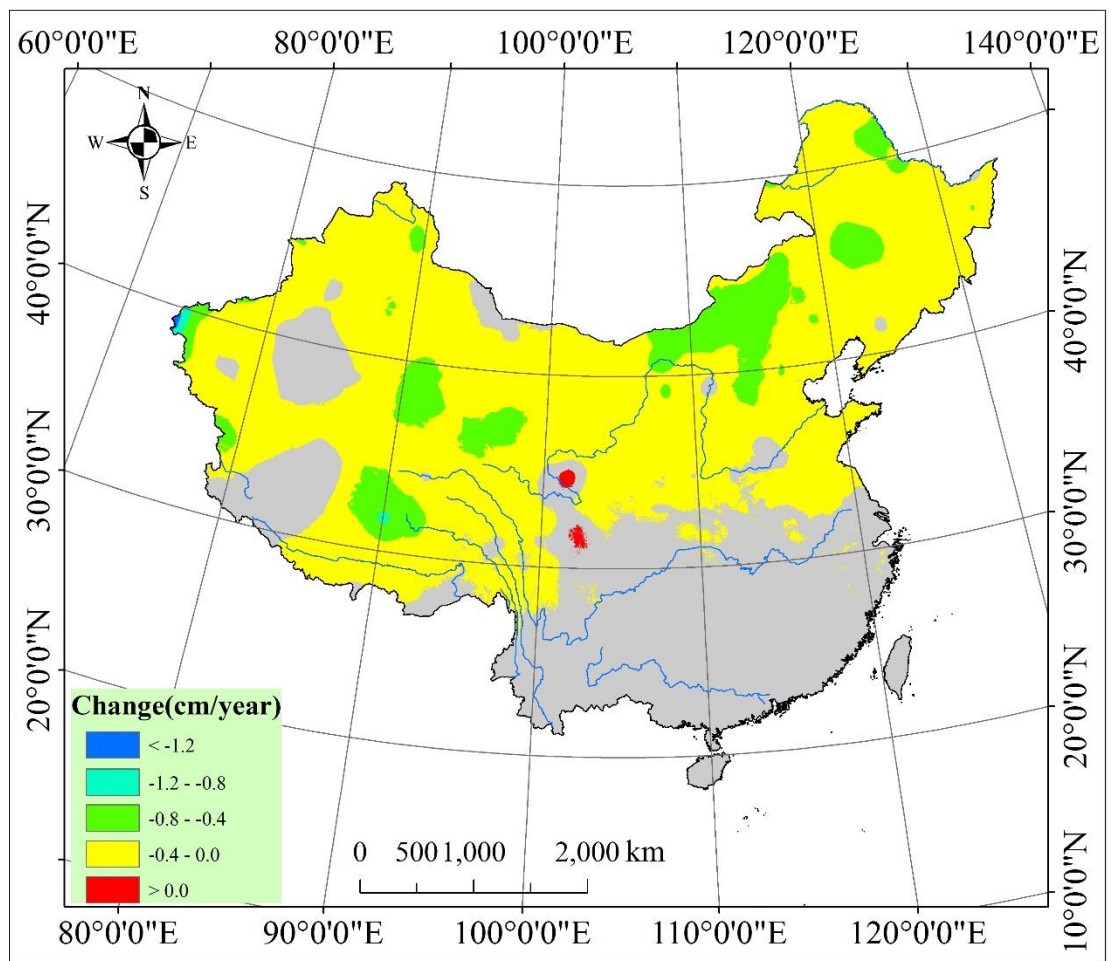

**Figure 9.** SFD trends across China from 1950 to 2009. The grey regions indicate non-significant
SFD changes, while trends in all other regions are statistically significant.

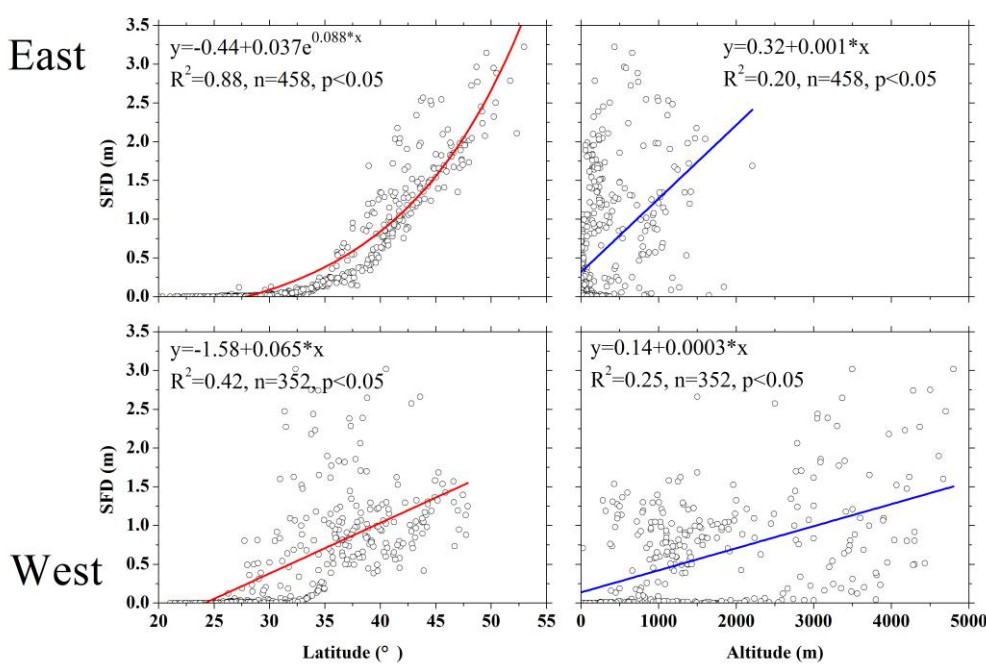

**Figure 10.** The relationship between SFD, latitude, and elevation in the east and west of China.

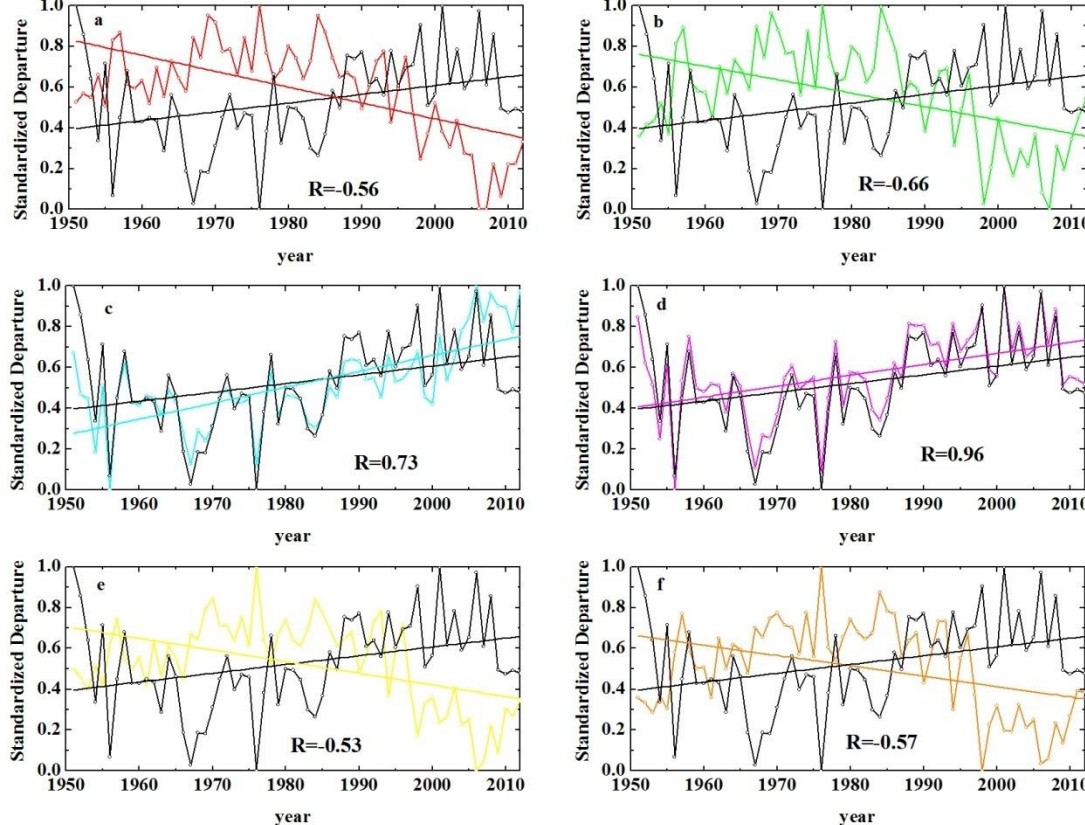

**Figure 11.** SFD time series and trend (black) and the potential forcing variables: (a) mean annual
ground surface temperature (red), (b) mean annual air temperature (green), (c) surface freeze
index (cyan), (d) air freezing index (magenta), (e) surface thawing index (yellow), (f) air thawing
index (orange). All variables are standardized to range from 0–1. R is the correlation coefficient,
and all are statistically significant.

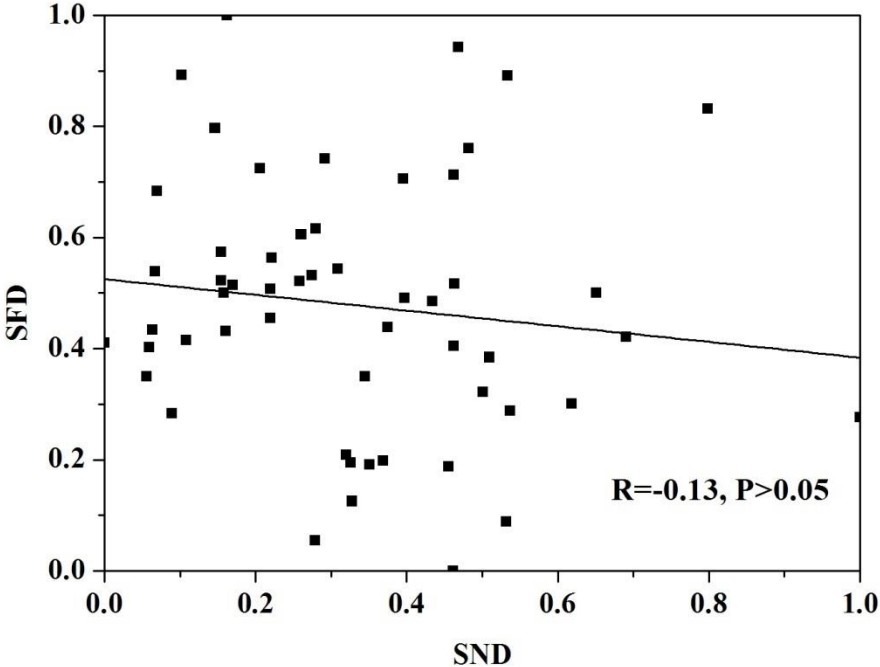

**Figure 12.** Correlation between SFD and SND. The variables are standardized to range from 0–1.

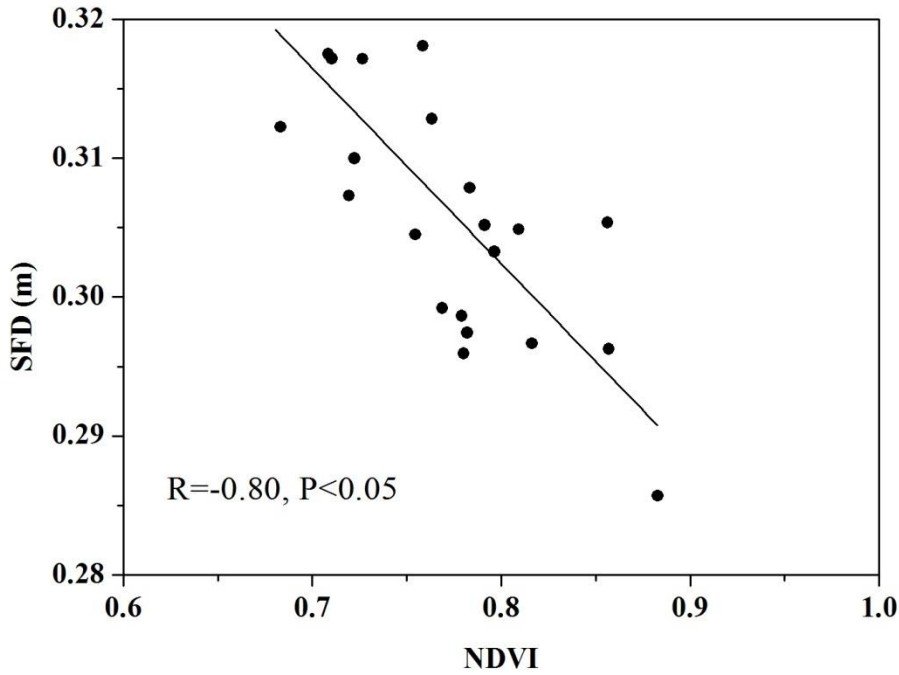

**Figure 13.** Correlation between SFD and mean annual NDVI.

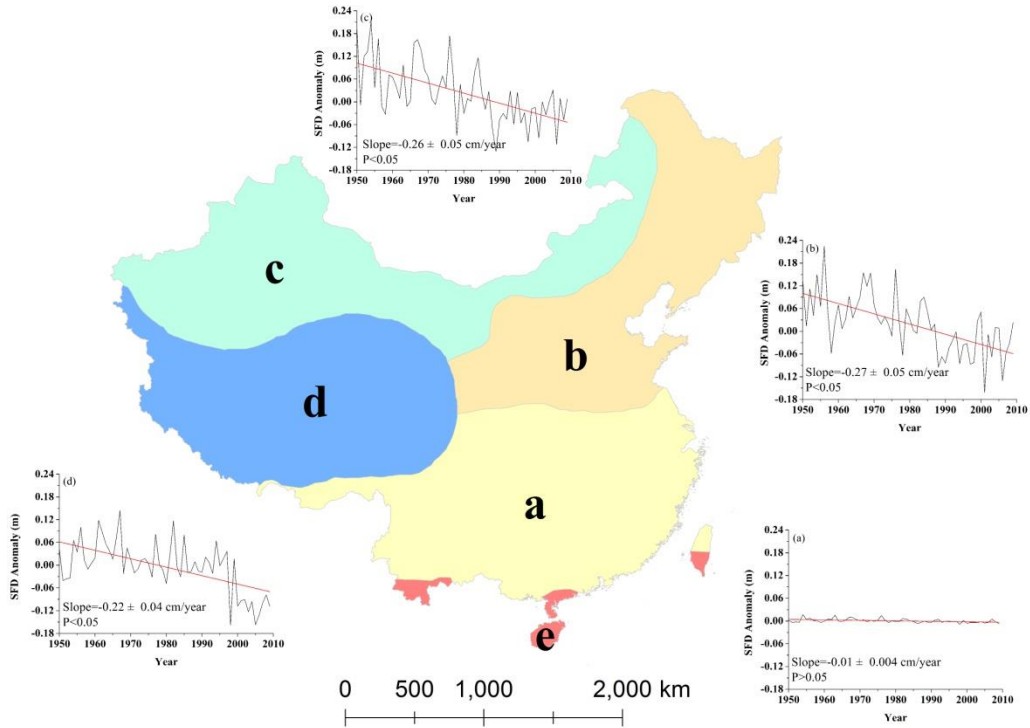

**Figure 14.** Time-series of SFD changes in different climate zones: (a) subtropical monsoon, (b)
temperate monsoon, (c) temperate continent, (d) Qinghai-Tibetan Plateau Alpine, and (e) tropical.
The insets are the SFD changes in the four climate zones; the bold black line is SFD, and bold red
line is the trend.