# Peer review of "Response of seasonal soil freeze depth to climate change across China"

_The Cryosphere, 2016_

## Referee Comment (RC1) · E. Jafarov (Referee) · 9 Nov 2016

This study focuses on spatial variations of seasonal soil freeze depth (SFD) and understanding of the environmental (non-climatic) factors contributing to the SFD variability and development in respond to climate change in China. I would suggest to change the 'non-climatic factors' to 'environmental factors' everywhere in the manuscript. This study addresses the importance of SFD as an indicator of climate change. The current version of the paper requires further improvements in language, flow and science. I enjoyed the discussion on snow depth and vegetation and was disappointed that discussion on soil moisture and soil organic matter relationship with SFD was missing. Using the fact that soil moisture and amount of organic matter play a significant role, in particular in Tibetan Plateau, it will be extremely important to include these environmental factors as well. Please find more detailed suggestion on how this paper could be improved. I would recommend this paper for the publication when suggested changes will be applied to the manuscript.

**1. Introduction**

L32. Out of 24% of permafrost affected soils in Northern Hemisphere how many falls in Tibetan Plateau? How much of the area in Tibetan Plateau permafrost-affected (i.e. has an active layer) and how much is seasonally frozen ground (i.e. no permafrost)?

L56. …feedbacks to climate change… please cite Abbott et al., (2016).

**2. Data and Methods**

Overall the flow in the Data and Methods section needs to be improved. Authors used different datasets (air temp., DEM, snow, etc). All these datasets are in different spatial resolutions. It is not clear to me what was the resolution of the final product and how authors dealt with all these different resolutions.

L78-80. Provide a web-link (reference) to the CMA dataset.

L95-103. Is it possible to divide the entire domain to several classes (subregions) with somewhat similar temperatures?

L104-107. DEM is finer resolution than MMGAT. Was it extrapolated to 0.5 deg or 0.5 deg was interpolated? Please clarify.

L109-112. Is the snow depth (SD) dataset available online? Provide a reference. More description is required. What is an overall snow distribution/max depth? How the SD was extrapolated? How accurate is that extrapolation (include uncertainties)? Where there more snow where is less? It would ice to know how this SD compares with MODIS or Globsnow products?

NDVI. Provide a description similarly to the SD (see previous comment). Note, the Resolution is 8km. How it was used (extrapolated interpolated)?

L118-120. Provide an uncertainty number associated with the interpolation.

L124-125. Did you use 2 DEM datasets? Previously it was 30m, here 1 km?

L131. More than 800 sites should go to the description of the monthly gridded air temperature

L137. Need to improve flow and rearrange. Repeating the met. station description. Move this sentence to the 2.1.3.

L141-143. Snow description should be moved to the 2.1.5. What about snow thermal properties?

L148. Provide a formula for $FI_a$.

L149. What dataset was used for soil thermal conductivities? Are they all constant and a one number or they change based on region and soil type?

L152. Similarly, to conductivities, which dataset was used for soil moisture content?

L154-156. Complex sentence. I suggest to simplify it or split to two.

L162. This is an important factor that has been referred in afterwards in the paper. State more clearly why it is important and what it tell us about SFD or permafrost.

L162-211. Make a workflow chart and refer to it while describing the workflow.

L205. …Stefan method – reference the equation.

**Results.**

L215. …Figure 4. What are the corresponding uncertainties?

L232. By calculating the anomalies for the whole region you average a lot of data. That is why it would be nice to divide the region on several classes, as I suggested above, and calculate anomalies for each subregion separately.

3.2. Section. Again within the subregions it could be easier to quantify changes in spatial variability.

L263-264. Belongs to discussion. Results section should only include the results description.

**Discussion**

L310—311. Rewrite that sense.

L314. What are pros and cons of using Stefan method?

L317-319 Combine two sentences for better flow.

L336. I would not say that TI can influence the ground temperature, because TI is an indicator rather than a factor.

L340. "snow structure" do you mean snow metamorphism?

L343. Authors did not find any relationship between SND and SFD. This confirms other findings, similarly Jafarov and Schaefer (2016) did not find any correlation between SND and ALT.

L344. Do not need 'the' before snow depth.

L344-350. Consider drawing plots of thermal offset (T_surface-T_air) vs. SND. This could reveal the relationship between offset and snow depth.

L351. Consider adding a plot of negative correlation of NDVI and SFD.

L355. …via different physical mechanism. Which mechanism?

**Conclusion**

Some of the conclusion could be moved to the abstract, especially statistics. If you divide your domain on subregions then you could better quantify the variability, and departures from mean in each subregion. This could improve the conclusion. Also which of the non-climatic or environmental factor influences SFD at most, and which one influences at least? Bringing in the influence of soil moisture and organic matter could strengthen the overall message.

**Figures**

Figure 2. Why the linear relationship was chosen? It looks quadratic or exponential to me.

Figure 3 c and d. Change y axis 'station number' to 'number of stations'

Also it would nice to see the relationship between SFD and elevation and SFD and latitude.

Figure 6 spatial variability. I suggest to choose different color bar (BlueWhiteRed), where 0 is blue, white in the middle, and red is 4.5m. This should improve the contrast and make it visually easier to understand.

Figure 7 is that the rate of change or a total change?

Figure 8. If the goal is to show the correlation, consider plotting SFD vs. MAGST and then MAAT, and so on.

Figure 9 and 10. Change y axis 'station number' to 'total number of stations'. Similarly, to Fig. 8 consider SFD vs. SND and NDVI and then you can colormap those points that will have the best correlation and analyze which years are those.

**References**

B.W. Abbott, J.B. Jones, E.A.G. Schuur, III, F.S.C., W.B. Bowden, M.S. Bret-Harte, H.E. Epstein, M.D. Flannigan, T.K. Harms, T.N. Hollingsworth, M.C. Mack, A.D. McGuire, S.M. Natali, A.V. Rocha, S.E. Tank, M.R. Turetsky, J.E. Vonk, K.P. Wickland, G.R. Aiken, H.D. Alexander, R.M.W. Amon, B.W. Benscoter, Y. Bergeron, K. Bishop, O. Blarquez, B. Bond-Lamberty, A.L. Breen, I. Buffam, Y. Cai, C. Carcaillet, S.K. Carey, J.M. Chen, H.Y.H. Chen, T.R. Christensen, L.W. Cooper, J.H.C. Cornelissen, W.J.d. Groot, T.H. DeLuca, E. Dorrepaal, N. Fetcher, J.C. Finlay, B.C. Forbes, N.H.F. French, S. Gauthier, M.P. Girardin, S.J. Goetz, J.G. Goldammer, L. Gough, P. Grogan, L. Guo, P.E. Higuera, L. Hinzman, F.S. Hu, G. Hugelius, E.E. Jafarov, R. Jandt, J.F. Johnstone, J. Karlsson, E.S. Kasischke, G. Kattner, R. Kelly, F. Keuper, G.W. Kling, P. Kortelainen, J. Kouki, P. Kuhry, H. Laudon, I. Laurion, R.W. Macdonald, P.J. Mann, P.J. Martikainen, J.W. McClelland, U. Molau, S.F. Oberbauer, D. Olefeldt, D. Paré, M.-A. Parisien, S. Payette, C. Peng, O.S. Pokrovsky, E.B. Rastetter, P.A. Raymond, M.K. Raynolds, G. Rein, J.F. Reynolds, M. Robard, B.M. Rogers, C. Schädel, K. Schaefer, I.K. Schmidt, A. Shvidenko, J. Sky, R.G.M. Spencer, G. Starr, R.G. Striegl, R. Teisserenc, L.J. Tranvik, T. Virtanen, J.M. Welker, S. Zimov. Biomass offsets little or none of permafrost carbon release from soils, streams, and wildfire: an expert assessment. Environ. Res. Lett., 11 (2016), p. 34014 http://dx.doi.org/10.1088/1748-9326/11/3/034014

Jafarov, E. and Schaefer, K.: The importance of a surface organic layer in simulating permafrost thermal and carbon dynamics, The Cryosphere, 10, 465-475, doi:10.5194/tc-10-465-2016, 2016.

---

## Referee Comment (RC2) · Anonymous Referee #2 · 10 Feb 2017

This is a valuable contribution, discussing soil freeze depth and its trends over China over several decades. The study is certainly worth being published in TC after the following comments have been considered.

You don't mention your paper "Response of changes in seasonal soil freeze/thaw state to climate change from 1950 to 2010 across China" in JGR, 121(11), pp.1984-2000, 2016. You should make very clear the differences to this paper and compare in detail the results and conclusions, i.e. build this paper on the previous one.

Check for typos and grammar!

Line 35: how can permafrost area (23%) and seasonally frozen ground (>80%) be more than 100%?

[Figure]

Section 2.1.3. Mention/discuss why no reanalysis data sets were used instead of MM-GAT? Reanalysis data could allow for some additional/alternative tests of meteorological parameters and their trends.

Section 2.16. What about other potentially important environmetal data (geology, wetness, other meteorological data, albedo, cloud cover . . .)? See also previous comment. It is not obvious why NDVI should be the most important other influence to SFD.

Line 128: wouldn't the usage of a reference level other than sea level, i.e a level closer to the real elevations (for instance, mean elevation of regions) be less sensitive to uncertainties in the estimated lapse rates? In particular for the Tibet Plateau, where most of the SFDs > 0 are found? Uncertainties would not be extrapolated but only interpolated.

Line 227: You list a number of reasons for the spatial SFD variability, but given no indication that they in fact could lead to the observed variations. Some influences, such as albedo, could actually be tested.

Mention and discuss the relation of soil freezing and permafrost from your data, as you mention permafrost at several places.

Fig. 1, 4a, 7: what is the inset to the lower right? It does not contribute. Remove.

Fig 4: your panel sequence is a, c, b, d? Why not a, b, c, d?

Fig 6: very hard to see differences. Better show anomalies with respect to the mean SFD?

Fig 10: I think the relation between SFD and NDVI needs more discussion. Why is it correlated on a year to year basis? You mainly discuss influences of vegatation on SFD, but couldn't both SFD and NDVI variations simply reflect the same drivers? Temperature -> Growing season? Temperature/precipitaion -> Water availability? I think it doesn't hold to just say Âń. . .the detailed physical mechanism will require further future workÂż You need to discuss at least the fundamental mechanisms, otherwise

showing the NDVI doesn't make much sense.

-

---

## Author Response (AR1)

Dear Dr. Kääb and Dr. Gruber,

Thank you very much for providing us the opportunity to improver our paper based on the referees' valuable comments. We have revised the manuscript according to the reviewers' comments and suggestions. Enclosed please find the revised manuscript, responses to the referees, as well as a list of changes. The responses are marked blue. We hope these revisions have improved our manuscript to make it suitable for publication in "The Cryosphere." If you have any questions or concerns about this paper, please don't hesitate to let me know. We look forward to hearing from you soon.

Sincerely yours,

Tingjun Zhang

The summary of the changes and responses to Referees' comments are listed below. The page, line, and figure numbers refer to our revised manuscript. The changes have been indicated in the paper using bold font.

We thank Dr. Elchin Jafarov for his detailed and insightful review of the discussion paper. We have addressed all the comments and made the suggested changes in the revised version of our manuscript. Our point-by-point replies (in blue) to the comments are listed below.

1. I would suggest to change the 'non-climatic factors' to 'environmental factors' everywhere in the manuscript.
Response: Thank you for this suggestion. We have changed 'non-climatic factors' to 'environmental factors' everywhere in the manuscript.

2. The current version of the paper requires further improvements in language, flow and science.
Response: We have revised the language and further improved the grammar and wording throughout the manuscript.

3. I enjoyed the discussion on snow depth and vegetation and was disappointed that discussion on soil moisture and soil organic matter relationship with SFD was missing.
Response: Thanks for this important reminder. We have explained these additional variables in detail in question 38 (3). Regarding soil moisture and organic matter: we agree that soil moisture and organic matter are two important factors influencing soil freeze depth. Based on previous research, soil moisture and organic matter have significant impacts on active layer thickness (Zhang et al., 1998, Jafarov and Schaefer, 2016). However, we analyze the soil freeze depth based on observational data, and unfortunately we cannot get observational soil moisture and organic matter datasets. If we used remote sensing data or analyzed products, there are many potential errors, e.g., remotely sensed soil moisture can only "see" the top 0–10 cm, the resolution of

the data is coarse, the accuracy is not good, etc.

4. L32. Out of 24% of permafrost affected soils in Northern Hemisphere how many falls in Tibetan Plateau? How much of the area in Tibetan Plateau permafrost-affected (i.e. has an active layer) and how much is seasonally frozen ground (i.e. no permafrost)?

Response: The latest estimates show that permafrost, seasonally frozen ground, and unfrozen ground cover areas of $1.06 \times 106$ km2 (40% of the Tibetan Plateau), $1.46 \times 106$ km2 (56% of the Tibetan Plateau), and $0.03 \times 106$ km2 (1% of the Tibetan Plateau), respectively, excluding glaciers and lakes which cover the remaining 3% (Zou et al., 2016). Permafrost regions occupy about $22.79 \times 10^6$ km$^2$ (approximately 24%) of the exposed land surface of the Northern Hemisphere (Zhang et al., 2003). Thus, combing with these dataset, permafrost of Tibetan Plateau occupies about 4.65% of the permafrost of Northern Hemisphere.

5. L56. …feedbacks to climate change… please cite Abbott et al., (2016).

Response: Thank you, we have added this new reference. Please see L62.

6. Overall the flow in the Data and Methods section needs to be improved. Authors used different datasets (air temp., DEM, snow, etc). All these datasets are in different spatial resolutions. It is not clear to me what was the resolution of the final product and how authors dealt with all these different resolutions.

Response: Thank you! We agree. We reformulated this part to clarify. Daily air temperature, daily ground temperature, daily soil temperature, NDVI, snow depth, mean monthly gridded air temperature (MMGAT), DEM, datasets are used in this study. Generally:

(1) Daily air and ground temperature data are used to estimate air/ground freezing/thawing index, and mean annual air temperature (MAAT), and mean annual ground surface temperature (MAGST) at the point scale;

(2) Daily soil temperature is used to estimate soil freeze depth in the point scale;

(3) Snow depth data are used to correlate soil freeze depth and snow depth in point scale.

(4) DEM is with 1-km spatial resolution, which is helpful for improve the MMGAT resolution and accuracy.

(5) Mean monthly gridded air temperature (MMGAT) data with $0.5° \times 0.5°$ resolution is combined with the 1-km resolution DEM and monthly lapse rates to obtain a 1 km $\times$ 1 km temperature dataset (the detail process can be seen in L152-159), which is then used to obtain soil freeze depth in the regional scale with 1-km resolution.

(6) NDVI data is with 8-km spatial resolution, which is used to analyze the relationship with soil freeze depth in the point scale. We obtained NDVI value based on the latitude and longitude information from stations.

7. L78-80. Provide a web-link (reference) to the CMA dataset.

Response: ok, we added the website in L96.

8. L95-103. Is it possible to divide the entire domain to several classes (subregions) with somewhat similar temperatures?

Response: Thanks, this is a good suggestion. The datasets used in the manuscript are just to estimate the freezing index. We worry that if we divide the entire domain to several subregions based on our subjective ideas, it may result in uncertainties or errors because we did not find any existing. Further, this was not one of the goals of this manuscript but we think that this would be a great idea for a future collaboration, perhaps based on some objective criteria and multivariate methodologies (PCA, clustering).

9. L104-107. DEM is finer resolution than MMGAT. Was it extrapolated to 0.5 deg or 0.5 deg was interpolated? Please clarify.

Response: MMGAT was interpolated to the DEM's finer resolution. More detail can be found in L150-159: To improve the original $0.5° \times 0.5°$ MMGAT data to a 1-km resolution, spatial interpolation was used in conjunction with monthly lapse rates and the 1-km resolution DEM (e.g., Willmott and Matsuura, 1995; Gruber et al., 2012). The data processing steps are to (1) calculate the average monthly atmospheric lapse rate based on all available meteorological stations across China and their elevations; (2) bring each average monthly gridded air temperature value to a reference level (elevation of 0 m) using the average monthly lapse rate; (3) apply a Kriging interpolation to the reference-level adjusted MMGAT; and (4) bring the gridded reference-level air temperature back to the DEM-gridded height. Based on more than 800 sites, we evaluated the interpolated MMGAT against the observational monthly air temperatures, and find that the regression coefficient is almost 1.0 with a minimum of 0.98 in April.

10. L109-112. Is the snow depth (SD) dataset available online? Provide a reference. More description is required. What is an overall snow distribution/max depth? How the SD was extrapolated? How accurate is that extrapolation (include uncertainties)? Where there more snow where is less? It would ice to know how this SD compares with MODIS or Globsnow products?

Response: The daily snow depth dataset is unfortunately not available publicly online, but we have added a reference in L128. In this study, we estimated the annual maximum snow depth based on the methodology described in L131–134. We only used daily snow depth observations, but no remotely sensed data. Because the goals of this study did not include the comparison between the observational snow depth and MOGIS or Globsnow products, we did not undertake this comparison.

11. NDVI. Provide a description similarly to the SD (see previous comment). Note, the Resolution is 8km. How it was used (extrapolated interpolated)?

Response: We added some discussion in the methods section on L139-140: "These data were used to assess the influence of vegetation on soil freeze depth. We extracted the NDVI values corresponding to the stations' latitude and longitude coordinates."

12. L118-120. Provide an uncertainty number associated with the interpolation.

Response: We revised this paragraph as follows (L142–149): "Missing data often present a potential problem for analyzing and averaging time series. Therefore, if fewer than five days were missing in a given month, filling in missing daily air temperatures was based on highly correlated neighboring sites using linear regression. Missing daily mean ground surface temperatures were estimated through linear regression with the daily mean air temperature at the same station. Based on the daily air temperature, we also calculate the mean monthly air temperature and mean annual air temperature (MAAT). The interpolated results are strongly correlated with observations, as indicated by regression coefficients larger than 0.95."

13. L124-125. Did you use 2 DEM datasets? Previously it was 30m, here 1 km?

Response: We used the GTOPO30 DEM data with a resolution of 30 arc-seconds, which is equal to 1 km. We revised this paragraph, please see L120-126: "Considering the complex terrain across China and the impacts of elevation on air temperature, we also used the global 30 arc-second elevation dataset (GTOPO30; https://lta.cr.usgs.gov/GTOPO30) as the digital elevation model (DEM) for this study to further improve the MMGAT resolution. GTOPO30 was derived from several raster and vector sources of topographic information. Across China, the elevation ranges from −152 to 8752 m (Figure 1). Based on this DEM, we spatially interpolate the MMGAT data to the DEM's 30 arc-second (1-km) resolution."

14. L131. More than 800 sites should go to the description of the monthly gridded air temperature

Response: Thanks! Yes. In this study, more than 800 sites of daily air temperature are used to evaluate the monthly gridded air temperature. Please see the L157-159 "Based on more than 800 sites, we evaluated the interpolated MMGAT against the observational monthly air temperatures, and find that the regression coefficient is almost 1.0 with a minimum of 0.98 in April.".

15. L137. Need to improve flow and rearrange. Repeating the met. station description. Move this sentence to the 2.1.3.

Response: Thanks! Based on your suggestions, we rewrote the "Data and Methods" section. We hope it is better now with an improved flow.

16. L141-143. Snow description should be moved to the 2.1.5. What about snow thermal properties?

Response: Thank you, we have moved it to section 2.1.5. We did not consider the snow thermal properties in our study, because we do not have this dataset.

17. L148. Provide a formula for FIa.

Response: We have added the formula in the methods section L160-174.

18. L149. What dataset was used for soil thermal conductivities? Are they all constant and a one number or they change based on region and soil type?

Response: In this study, we used the simple Stefan Solution, which applies the edaphic factor, representing soil thermal conductivity, soil bulk density, water content, and latent heat of fusion. Thus, the soil thermal conductivity is different at each station. Further, it is difficult to obtain the soil thermal conductivity at the regional scale. The simple Stefan Solution can be efficient in applying estimated soil freeze depth at the regional scale.

19. L152. Similarly, to conductivities, which dataset was used for soil moisture content?

Response: As above regarding question 18, the edaphic factor included soil moisture content as part of the simple Stefan Solution.

20. L154-156. Complex sentence. I suggest to simplify it or split to two.

Response: Thank you, we split the sentences into two, please see L185-188: "However, based on the SFD and annual freezing index at each observational site, we can quantify the relationship between these two parameters (Figure 2). We find a strong and statistically significant correlation of R=0.87."

21. L162. This is an important factor that has been referred in afterwards in the paper. State more clearly why it is important and what it tell us about SFD or permafrost.

Response: We describe the importance of soil freeze depth across China in the introduction, please see L46-75. For SFD or permafrost, please see L107-109 where we explain that "we combine the potential maximum soil seasonal freeze depth in permafrost regions, and maximum soil freeze depth in SFG."

22. L162-211. Make a workflow chart and refer to it while describing the workflow.

Response: Great suggestion—we made such a workflow chart, introducing how to estimate soil freeze depth using Stefan solution (Fig. 1):

[Figure]

Fig. 1. Flowchart describing how to estimate SFD at the regional scale.

23. L205. …Stefan method – reference the equation.
Response: We added equation 4 in L190.

24. L215. …Figure 4. What are the corresponding uncertainties?
Response: As an estimate of the uncertainty of SFD, we provide the standard deviation at each site across China (Fig. 2 below), please see L222-227 "Figure 5 shows the standard deviation of SFD at each site across China. It varies from 0.00–0.27 m. The standard deviation of SFD is generally less than 0.03 m south of 35 N, except on the Tibetan Plateau. In northeastern China, the standard deviation changes between 0.06 m and 0.15 m. In the northwest, it is generally 0.06–0.12 m. On the Tibetan Plateau, the standard deviation varies from less than 0.09 m, but can be greater than 0.18 m at some sites."

Another SFD uncertainties is the comparison between observational SFD and simulated SFD, Fig. 3 (in the manuscript) have shown it.

[Figure]

Fig. 2. The standard deviation of SFD across China.

25. L232. By calculating the anomalies for the whole region you average a lot of data. That is why it would be nice to divide the region on several classes, as I suggested above, and calculate anomalies for each subregion separately.

Response: Please see our response below, regarding your comment #38. While it would be interesting to calculate the anomalies of SFD in subregions, there would be different numbers of observational sites in each subregion. Further, we can see the SFD trend in different subregions in Fig. 4c. Here, we just want to provide the time-series of SFD trend in China as a whole, because this has not previously been done and represents a new result.

For the sub-regional averaged SFD changes in different sub-regions, we have added it in discussion part 4.2. Please see the detail explanation in L383-397.

26. 3.2. Section. Again within the subregions it could be easier to quantify changes in spatial variability.

Response: Thanks for your nice suggestion. We discussed the SFD changes in different sub-regions, based on climate zones standard. Please see the detail in L383-397.

27. L263-264. Belongs to discussion. Results section should only include the results description.

Response: Thanks! We add one sentence here, and discuss the relationship between

air temperature and SFD in the other sections. Please see L259-261 "Therefore, air temperature is possibly one of the important factors that influence soil freeze depth in these areas. More detailed discussion is provided in sections 3.3 and 4.1."

28. L310—311. Rewrite that sense.
Response: We rewrote this, "Similarly, soil freeze/thaw depth changes also have destabilizing effects on engineering structures, such as on improperly constructed infrastructure".

29. L314. What are pros and cons of using Stefan method?
Response: Soil freeze depth are affected by so many variables, such as soil moisture, soil density, soil texture, thermal conductivity etc. It is difficult or impossible to obtain these parameters in the regional scale. While simple Stefan method provide a catch-all E factor to representing these parameters as a whole. Thus, it is advantage estimating soil freeze depth in the regional scale. Further, this method is successfully using the related study (Zhang et al., 2005; Park et al., 2016).
    Except these advantages, some disadvantages are also exist. Due to the catch-all E factor including so many parameters, it limit the accuracy of soil freeze depth comparing with observational dataset. But the uncertainties can be accepted by us (question 24).

30. L317-319 Combine two sentences for better flow.
Response: Thanks! We combined into one sentences, see L315-317 "SFD variability is susceptible to climate warming and environmental change, and is affected by variables including air temperature, ground surface temperature, freezing/thawing index, and vegetation.".

31. L336. I would not say that TI can influence the ground temperature, because TI is an indicator rather than a factor.
Response: Good point, we revised it as follows in L334: "Thus TI is a potential indicator of SFD, indirectly affecting soil temperature"

32. L340. "snow structure" do you mean snow metamorphism?
Response: "snow structure" is the terminology from Park et al. (2015). Based on the reference, it means snow cover, snow depth, and snow density. However, snow structure can include more properties, such as snow type (Zhong et al., 2014), snow days, snow water equivalent, grain size, and so on.

33. L343. Authors did not find any relationship between SND and SFD. This confirms other findings, similarly Jafarov and Schaefer (2016) did not find any correlation between SND and ALT.
Response: Yes, we agree; in this study, the relationship between SFD and SND is negative, but not statistically significant. Thus, we further analyze the possible reasons in L336-349. Jafarov and Schaefer (2016) 's result make us more confidence

for this result. We have cited this reference in this study.

34. L344. Do not need 'the' before snow depth.
Response: Thanks! We delete it.

35. L344-350. Consider drawing plots of thermal offset (T_surface-T_air) vs. SND. This could reveal the relationship between offset and snow depth.
Response: We estimated the relationship between snow depth and thermal offset (T_surface − T_air) across China (Fig. 3 below). It shows that there is a statistically significant negative correlation between snow depth and thermal offset, though this relationship is mostly caused by some potential outliers. Although there is a slight negative correlation, we focus on soil freeze depth and snow depth in this study. Soil freeze depth is not only at the surface, but also below ground. It is thus more complex and needs further research.

[Figure]

Fig. 3. Correlation between snow depth and thermal offset (difference between ground surface temperature and air temperature).

36. L351. Consider adding a plot of negative correlation of NDVI and SFD.
Response: Thank you—the negative correlation between NDVI and SFD is in Fig. 11.

37. L355. …via different physical mechanism. Which mechanism?
Response: We apologize for this confusing sentence, and have revised it as follows (see L353-354):"….. via different physical mechanisms (Snyder et al., 2004), e.g. changes in the surface albedo……"

38. Some of the conclusion could be moved to the abstract, especially statistics. If you divide your domain on subregions then you could better quantify the variability, and departures from mean in each subregion. This could improve the conclusion. Also which of the non-climatic or environmental factor influences SFD at most, and which one influences at least? Bringing in the influence of soil moisture and organic matter could strengthen the overall message.

Response: Great suggestions—we added these conclusions and statistics to the abstract, please see L24–31: "Investigating potential climatic and environmental driving factors of soil freeze depth variablity, we find that mean annual air temperature and ground surface temperature, air thawing index, ground surface thawing index, and vegetation growth are all negatively associated with soil freeze depth. Changes in snow depth are not correlated with soil freeze depth. Air and ground surface freezing index are positively correlated with soil freeze depth. Comparing these potential driving factors of soil freeze depth, we find that freezing index and vegetation growth are more strongly correlated with soil freeze depth, while snow depth is not significant."

Sub-region problem: this is a very good suggestion for spatial analysis. Thus, based on the potential driving variables of SFD, we divide China into five climate zones, which is powerful used in the related study (Zheng et al., 2010). Please see L383-397. Hopefully, you can agree with us.

Regarding soil moisture and organic matter: we agree that soil moisture and organic matter are two important factors influencing soil freeze depth. Based on previous research, soil moisture and organic matter have significant impacts on active layer thickness (Zhang et al., 1998, Jafarov and Schaefer, 2016). However, we analyze the soil freeze depth based on observational data, and unfortunately we cannot get observational soil moisture and organic matter datasets. If we used remote sensing data or analyzed products, there are many potential errors, e.g., remotely sensed soil moisture can only "see" the top 0–10 cm, the resolution of the data is coarse, the accuracy is not good, etc.

39. Figure 2. Why the linear relationship was chosen? It looks quadratic or exponential to me.
Response: Based on previous studies (Nelson and Outcalt, 1987; Shiklomanov et al., 2002; Zhang et al., 2005; Park et al., 2016), the relationship between soil freeze depth and freezing index was suggested to be linear.

40. Figure 3 c and d. Change y axis 'station number' to 'number of stations'
Also it would nice to see the relationship between SFD and elevation and SFD and latitude.
Response: Thanks! We revised this as suggestged.
    For the relationship between SFD and elevation and latitude, we do have figures but we were not sure whether they should be included. We provide them below, and

could add them to the paper if you and the editors believe they add to our findings:
 "To explore the spatial features of SFD, we classify the meteorological stations as either eastern or western based on 110°E longitude. Figure 4 represents the correlations between SFD and latitude and altitude in the eastern and western parts. In the east, we find an exponential relationship between SFD and latitude, and a linear relationship with altitude, with both being statistically significant. The SFD value ranges from 0.0 m to less than 3.5 m, varying with latitude more so than with altitude. Thus, SFD was mainly affected by latitude in the east of China. In the west, SFD is near 0.0 m with altitude higher than 1000m, because these sites are located in the Yunnan-Guizhou Plateau, but with lower latitude. Similarly, SFD is related statistically significantly with altitude and latitude in west, where altitude was the main factor affecting SFD."

[Figure]

Fig. 4. The relationship between SFD and latitude and altitude in the east and west of China, as divided by 110°E longitude.

41. Figure 6 spatial variability. I suggest to choose different color bar (BlueWhiteRed), where 0 is blue, white in the middle, and red is 4.5m. This should improve the contrast and make it visually easier to understand.
Response: Thanks! We revised the color bar.

42. Figure 7 is that the rate of change or a total change?
Response: Figure 8 (original figure 7) shows the rate of SFD change during 1950-2010 across China.

43. Figure 8. If the goal is to show the correlation, consider plotting SFD vs. MAGST

and then MAAT, and so on.

Response: The goal of figure 9 (original figure 8) is not only for the correlation, but also to show the variability of SFD, MAGST, MAAT, etc.

44. Figure 9 and 10. Change y axis 'station number' to 'total number of stations'. Similarly, to Fig. 8 consider SFD vs. SND and NDVI and then you can colormap those points that will have the best correlation and analyze which years are those.

Response: ok, we revised these figures, please see new figures 10 and 11.

Reference:

1. Abbott B W, Jones J B, Schuur E A G, et al. Biomass offsets little or none of permafrost carbon release from soils, streams, and wildfire: an expert assessment[J]. Environmental Research Letters, 2016, 11(3): 034014.

2. Jafarov E, Schaefer K. The importance of a surface organic layer in simulating permafrost thermal and carbon dynamics[J]. The Cryosphere, 2016, 10(1): 465-475.

3. Nelson, F. E. and Outcalt, S. I.: A computational method for prediction and regionalization of permafrost, Arctic and Alpine Research, 279-288, doi: 10.2307/1551363, 1987.

4. Park H, Kim Y, Kimball J S. Widespread permafrost vulnerability and soil active layer increases over the high northern latitudes inferred from satellite remote sensing and process model assessments[J]. Remote Sensing of Environment, 2016, 175: 349-358.

5. Park H, Fedorov A N, Zheleznyak M N, et al. Effect of snow cover on pan-Arctic permafrost thermal regimes[J]. Climate Dynamics, 2015, 44(9-10): 2873-2895.

6. Peng X, Zhang T, Cao B, et al. Changes in freezing-thawing index and soil freeze depth over the Heihe River Basin, western China[J]. Arctic, Antarctic, and Alpine Research, 2016, 48(1): 161-176.

7. Shiklomanov N I. Non-climatic factors and long-term, continental-scale changes in seasonally frozen ground[J]. Environmental Research Letters, 2012, 7(1): 011003.

8. Shiklomanov N I, Nelson F E. Active‐layer mapping at regional scales: A 13‐year spatial time series for the Kuparuk region, north‐central Alaska[J]. Permafrost and Periglacial Processes, 2002, 13(3): 219-230.

9. Snyder, P., Delire, C., and Foley, J.: Evaluating the influence of different vegetation biomes on the global climate, Climate Dynamics, 23, 279-302, doi: 10.1007/s00382-004-0430-0, 2004.

10. Zhang T, Frauenfeld O W, Serreze M C, et al. Spatial and temporal variability in active layer thickness over the Russian Arctic drainage basin[J]. Journal of Geophysical Research: Atmospheres, 2005, 110(D16).

11. Zhang T, Stamnes K. Impact of climatic factors on the active layer and permafrost at Barrow, Alaska[J]. Permafrost and Periglacial Processes, 1998, 9(3): 229-246.

12. Zhang T, Barry R G, Knowles K, et al. Distribution of seasonally and perennially

frozen ground in the Northern Hemisphere[C]//Proceedings of the 8th International Conference on Permafrost. AA Balkema Publishers, 2003, 2: 1289-1294.

13. Zheng J, Yin Y, Li B. A New Scheme for Climate Regionalization in China [J]. ACTA GEOGRAPHICA SINICA, 2010, 65(1): 3-12.

14. Zhong X, Zhang T, Wang K. Snow density climatology across the former USSR[J]. The Cryosphere, 2014, 8(2): 785-799.

15. Zou, D., Zhao, L., Sheng, Y., Chen, J., Hu, G., Wu, T., Wu, J., Xie, C., Wu, X., Pang, Q., Wang, W., Du, E., Li, W., Liu, G., Li, J., Qin, Y., Qiao, Y., Wang, Z., Shi, J., and Cheng, G.: A New Map of the Permafrost Distribution on the Tibetan Plateau, The Cryosphere Discuss., doi:10.5194/tc-2016-187, in review, 2016.

We thank the Anonymous Referee's for their comments on our manuscript. We also appreciate the careful consideration and detailed evaluation. Our replies are included in blue font.

1. You don't mention your paper "Response of changes in seasonal soil freeze/thaw state to climate change from 1950 to 2010 across China" in JGR, 121(11), pp.1984-2000, 2016. You should make very clear the differences to this paper and compare in detail the results and conclusions, i.e. build this paper on the previous one.

Response: Thank you very much for giving us the opportunity to clarify the differences between this manuscript submitted to The Cryosphere, and our study already published in JGR. The unique aspects of this submitted manuscript are that, rather than using stations records alone and focusing on the point-scale or using coarsely gridded data at the large scale, we develop regional-scale gridded fields based on the combination of station data and gridded data. We then employ the Stefan solution to investigate soil freeze depth at the point and regional scale. The JGR manuscript did not do this. Furthermore, we also analyze the potential driving factors (including climate and environmental factors), which, again, JGR publication did not do. Please see the difference in Table 1. In the study, we also introduced the difference, please see L69-72.

Table 1. Comparison of two manuscripts

|  | *The Cryosphere* Manuscript | *JGR-Earth Surface* Manuscript |
|---|---|---|
| Study Target | Soil freeze depth (SFD). | surface soil freeze/thaw status |
| Objectives | 1) To investigate the spatiotemporal variability of seasonal soil freeze depth from the point- to the regional-scale
2) To analyze the potential forcing variables of soil freeze depth across China. | 1) To assess the spatiotemporal variation of seasonal soil freeze/thaw status across China, incorporating a land cover classification |
| Methodology | 1) Using daily air temperature to estimate freezing index, and daily soil temperature to compute the soil freeze depth, we obtain the edaphic factor (E-factor) through the simplified Stefan Solution
2) Combining freezing index, derived from gridded air temperature, and the E-factor, we estimate the spatiotemporal variability of | 1) Establish the relationship between monthly air temperature and monthly freeze days (based on soil temperature at 5 cm) in different land cover types, use monthly air temperature ranges in each land cover type to classify the surface soil freeze/thaw states into three types: completely frozen (CF), partially frozen (PF), or |

| | | |
|---|---|---|
| | soil freeze depth at the regional scale using the simplified Stefan Solution | unfrozen (UF)
2) Use gridded monthly air temperature to quantify the spatial variability of soil freeze/thaw states, and evaluate the area extent of surface soil freeze/thaw states at the monthly and annual scale |
| Main Conclusions | 1) The spatial distribution of SFD variability is influenced by latitude and elevation across China;
2) Using 839 sites we found that the SFD decreased significantly, at -0.18 cm/year from 1967 to 2012, equal to a net change of 8.05 cm;
3) On the regional scale, the 1950–2009 spatial variation of SFD ranges 0.0–4.5 m across China, with most areas exhibiting significant decreases between less than 0.0 and -0.4 cm/year;
4) A negative between SFD and mean annual air temperature (MAAT), mean annual ground surface temperature (MAGST), $TI_a$ (air thawing index), and $TI_s$ (surface thawing index). Surprisingly, we found that there is no correlation between SFD and SND. The environmental factor vegetation (NDVI) is negatively correlated with SFD, indicating that 64% of the changes in SFD can be accounted for by vegetation. | 1) Changes in area extent of seasonal soil freeze/thaw state are somewhat different and complicated compared to temperature trends. The mean annual area extent of soil CF state decreased statistically significantly at a rate of $-0.043 \times 10^6$ km$^2$/decade from 1950 to 2010. For the soil UF state, the mean annual area extent increased significantly by about $0.037 \times 10^6$ km$^2$/decade. However, the mean annual area extent of soil PF state increased statistically significantly by $0.032 \times 10^6$ km$^2$/decade from 1950 to 1993, and exhibited no change from 1993 to 2010.
2) The monthly area extent of CF state decreased significantly for all months, but decreased for the UF state. The PF state showed a complex pattern, increasing during November–March and decreasing in the other months.
3) During 1950–2010, the freeze status value decreased statistically significantly from winter to summer, and increased from spring to summer. Spatially, the maximum status value was |

| | | | mainly located in the south of China. The minimum value was in the north of China and on the Tibetan Plateau. |
|---|---|---|---|

2. Line 35: how can permafrost area (23%) and seasonally frozen ground (>80%) be more than 100%?

Response: Thank you for catching this. We have revised it in L42-44 "…. or approximately 23% of its land area, mainly on the Tibetan Plateau; regions with SFG occupy about 50% of the land area in China (Zhou et al., 2000)."

3. Section 2.1.3. Mention/discuss why no reanalysis data sets were used instead of MMGAT? Reanalysis data could allow for some additional/alternative tests of meteorological parameters and their trends.

Response: In this study, we chose the gridded observational dataset from the University of Delaware's 1900–2014 terrestrial air temperature gridded monthly time series. The reason is that this dataset combined the completely observation station data, considered the complex terrain, which have been used for frozen ground study across China (Peng et al., 2016). Considering the complex terrain of frozen ground distribution, we used a simple and popular method to improve the accuracy of gridded air temperature, and the detailed description of the method is in L150-159.

The MMGAT dataset has been evaluated against meteorological station data, and the result (please see L157-159) indicates good agreement.

4. Section 2.16. What about other potentially important environmental data (geology, wetness, other meteorological data, albedo, cloud cover …)? See also previous comment. It is not obvious why NDVI should be the most important other influence to SFD.

Response: Thank you for your suggestion. As you say, there are many other environmental factors affecting SFD. However, it is difficult or impossible to obtain these in-situ data, and some factors we cannot quantify. Thus, we chose some of the more obtainable variables such as NDVI. We did not think that NDVI would be the most important factor influencing SFD, but figured it might be important. Compared with other environment factors, NDVI is a relatively reliable product.

The reasons why we choose NDVI rather than other environmental variables, e.g. geology, wetness, albedo, cloud cover are:

(1) At the regional scale, NDVI is considered as more reliable by comparing with observational data (Bao et al., 2015). Taking wetness for an example, a reanalysis or remote sensing product would not have very good accuracy compared to observational data, especially in the cold seasons. Further, remote sensing can only get soil moisture in a shallow, to soil layer. (Yang et al., 2007; Chen et al., 2013). Therefore, we consider NDVI as the only environmental factor here.

(2) NDVI is selected here to partly represent the influences of soil conditions (e.g. wetness and soil type), topography (slope aspect), geology and albedo since NDVI can react these potential variables.

5. Line 128: wouldn't the usage of a reference level other than sea level, i.e a level closer to the real elevations (for instance, mean elevation of regions) be less sensitive to uncertainties in the estimated lapse rates? In particular for the Tibet Plateau, where most of the SFDs > 0 are found? Uncertainties would not be extrapolated but only interpolated.

Response: Thanks! We agree. For the sea level or a reference level question, we revised it as reference level in L154-155 "…. a reference level (elevation of 0 m)…".

For the uncertainties question, the reason why we used this method to process MMGAT is because of the complex terrain across China, especially in the mountain area in western of China. Through the 1-km DEM dataset and the lapse rates, we can improve the accuracy of MMGAT (Qin et al., 2015 & 2016; Zou et al., 2015).

6. Line 227: You list a number of reasons for the spatial SFD variability, but given no indication that they in fact could lead to the observed variations. Some influences, such as albedo, could actually be tested.

Response: Thank you for your suggestion. In the previous manuscript L227, it just list several possible reasons for SFD variability in northwest of China. Taking a panoramic view of the study, this part seems confused. Further, we have detail discussion about the potential driving factors of soil freeze depth. Please see the more comprehensive explanations about it in section 4.1. For your suggestion about albedo, at the regional scale, albedo product includes remote sensing dataset (e.g. MODIS, GLASS), and reanalysis datasets (e.g. ERA-Interim). These datasets are not good agreement with observational data, especially in the cold season, and with snow cover (Fig. 1). However, observational albedo data are really difficult to obtain. Your suggestions are really good, and we will further obtain the dataset in the field in a special study area, and hopefully can get new results for this study.

Hopefully, you can agree with us. Thanks!

[Figure]

Fig.1 Comparison of albedo between observational and others derived remote sensing dataset (GLASS), reanalysis dataset (ERA-Interim). Combined with in-situ albedo of less than 90 sites (only 20 sites across China), here evaluate the GLASS albedo with 8-day temporal resolution, and daily ERA-Interim albedo during1996-2012. The RMSE of albedo is 0.25 and 0.55, respectively. Results show that GLASS and ERA-Interim albedo are not good agreement with in-situ, which are not suitable for using in this study.

7. Mention and discuss the relation of soil freezing and permafrost from your data, as you mention permafrost at several places.

Response: Ok! Soil freeze depth in seasonally frozen ground represents the *maximum* soil freeze depth, while in permafrost regions it means *potential* soil freezing depth. However, we mostly focus on the maximum thaw depth (actually active layer thickness) in permafrost regions. In fact, the potential soil freezing depth in permafrost regions can also reflect climate change in permafrost regions (Zhou et al., 2000).

8. Fig. 1, 4a, 7: what is the inset to the lower right? It does not contribute. Remove.
Response: Thanks! We remove it.

9. Fig 4: your panel sequence is a, c, b, d? Why not a, b, c, d?
Response: Ok, we have revised it.

10. Fig 6: very hard to see differences. Better show anomalies with respect to the mean SFD?
Response: We agree, however, Fig. 7 (original figure 6) is intended to show the spatial distribution of SFD for several decades. Fig. 8 (original figure 7) can show the temporal variability of SFD. Thus, fig. 8 shows the differences. Below we show the SFD anomaly in decades with respect to the 1950–2009 mean. If you and the editors think this figure is useful to include in our manuscript, we can gladly do so.

[Figure]

Figure 2. Spatial variability of SFD anomaly for the decades of the 1950s, 1960s, 1970s, 1980s, 1990s, and 2000s, with respect to the 1950–2009 mean across China.

11. Fig 10: I think the relation between SFD and NDVI needs more discussion. Why is it correlated on a year to year basis? You mainly discuss influences of vegetation on SFD, but couldn't both SFD and NDVI variations simply reflect the same drivers? Temperature -> Growing season? Temperature/precipitation -> Water availability? I think it doesn't hold to just say….the detailed physical mechanism will require further future work. You need to discuss at least the fundamental mechanisms, otherwise showing the NDVI doesn't make much sense.

Response: Thank you for pointing this out. We have added more discussion about the relationship between SFD and NDVI in the discussion part. The reason why we analyzed the relationship between annual NDVI and SFD is that the SFD represents the maximum soil freeze depth in one year (i.e. it is an annual value).

For the relationship between SFD and NDVI, we have explained it in two ways, please see L350-378 "A negative correlation between SFD and vegetation, as quantified by NDVI, is found. Vegetation change has a significant influence on the climate system mostly through changes to the surface radiative energy budget, which can be affected the SFD. Based on previous research, vegetation varies in different land cover types and responds to climate change via different physical mechanisms (Snyder et al., 2004), e.g., changes in the surface albedo (e.g., bare ground versus vegetation cover), vegetation transpiration, and shading effects (Kelley et al., 2004; Snyder et al., 2004; Swann et al., 2010; Chang et al., 2012; Zhang et al., 2012). In the cold season, less/decreased vegetation will be more easily snow covered, thus increasing the albedo considerably. Increasing albedo results in less net radiation at the land surface, as more incoming solar radiation is reflected from the surface. Then, the surface air temperature will decrease considerably due to less energy absorbed at the surface. For the colder land surface, the sensible heat flux is reduced. Further, the vegetation decrease results in reducing evapotranspiration, which decreases the latent heat flux (Snyder et al., 2004). Compared to increased vegetation cover, less vegetation causes a large annual-average increase in the surface albedo with the largest changes in the winter and spring seasons, which reduces the amount of net radiation at the surface, making the surface colder and resulting in SFD increases. Conversely, vegetation increases could lead to decreasing SFD. The vegetation's effect on transpiration is primarily important in summer, while SFD primary occurs in winter and spring (Snyder et al., 2004).

The significant negative correlation between NDVI and SFD demonstrates their inverse relationship. Results from many previous studies indicated that there has been a vegetation increase, or a greening trend, in different regions during the past several decades (Peng et al., 2011; Piao et al., 2011; Zhang et al., 2013; Zhu et al., 2016). Because climate change controls the spatial distribution of vegetation, most studies examine vegetation variability as impacted by climate change, including temperature and precipitation (Bao et al., 2015; Huang et al., 2016). Results showed that increasing temperature and precipitation result in vegetation increases. Similarly, figure 8 shows that rising temperature results in a SFD decrease. The negative relationship between SFD and NDVI indicates the effect of vegetation on SFD, and

also their inverse relationship."

Reference:

1. Bao, G., Bao, Y., Sanjjava, A., Qin, Z., Zhou, Y., and Xu, G.: NDVI-indicated long-term vegetation dynamics in Mongolia and their response to climate change at biome scale, International Journal of Climatology, 35, 4293-4306, doi: 10.1002/joc.4286, 2015.

2. Chang, X. L., Jin, H. J., Wang, Y. P., Zhang, Y. L., Zhou, G. Y., Che, F. Q., and Zhao, Y. M.: Influences of vegetation on permafrost: A review, Acta Ecologica Sinica, 32, 7981-7990, doi: 10.5846/stxb201202120181, 2012.

3. Chen, Y., Yang, K., Qin, J., Zhao, L., Tang, W., & Han, M.: Evaluation of amsr-e retrievals and gldas simulations against observations of a soil moisture network on the central tibetan plateau.Journal of Geophysical Research Atmospheres, 118(10), 4466–4475, 2013.

4. Huang, F., Mo, X., Lin, Z., and Shi, H.: Dynamics and responses of vegetation to climatic variations in Ziya-Daqing basins, China, Chinese Geographical Science, 26, 478-494, doi: 10.1007/s11769-016-0807-0, 2016.

5. Kelley, A. M., Epstein, H. E., and Walker, D. A.: Role of vegetation and climate in permafrost active layer depth in arctic tundra of northern Alaska and Canada, Journal of Glaciology and Geocryology, 26, 269-274, 2004.

6. Peng, S., Chen, A., Xu, L., Cao, C., Fang, J., Myneni, R. B., Pinzon, J. E., Tucker, C. J., and Piao, S.: Recent change of vegetation growth trend in China, Environmental Research Letters, 6, 044027, doi: 10.1088/1748-9326/6/4/044027, 2011.

7.Peng, X., Frauenfeld, O. W., Cao, B., Wang, K., Wang, H., Su, H., Huang, Z., Yue, D., and Zhang, T.: Response of changes in seasonal soil freeze/thaw state to climate change from 1950 to 2010 across china, Journal of Geophysical Research: Earth S urface, 121, 1984–2000, doi: 10.1002/2016JF003876, 2016.

8. Piao, S., Wang, X., Ciais, P., Zhu, B., Wang, T., and Liu, J.: Changes in satellite-derived vegetation growth trend in temperate and boreal Eurasia from 1982 to 2006, Global Change Biology, 17, 3228–3239, doi: 10.1111/j.1365-2486.2011.02419.x, 2011.

9. Qin, Y., Tonghua, W. U., Ren, L. I., Xie, C., Zou, D., & Zhang, L., et al. : The applicability of era-interim land surface temperature dataset to map the permafrost distribution over the tibetan plateau. Journal of Glaciology & Geocryology, 2015.

10. Qin, Y., Tonghua, W. U., Ren, L. I., Xie, C., Qiao, Y., & Chen, H., et al.: Application of era product of land surface temperature in permafrost regions of qinghai-xizang plateau. Plateau Meteorology, 2015.

11. Snyder, P., Delire, C., and Foley, J.: Evaluating the influence of different vegetation biomes on the global climate, Climate Dynamics, 23, 279-302, doi: 10.1007/s00382-004-0430-0, 2004.

12. Swann, A. L., Fung, I. Y., Levis, S., Bonan, G. B., and Doney, S. C.: Changes in Arctic vegetation amplify high-latitude warming through the greenhouse effect, Proceedings of the National Academy of Sciences, 107, 1295-1300, doi:

10.1073/pnas.0913846107, 2010.

13. Yang, K., Watanabe, T., Koike, T., Xin, L. I., Fujii, H., & Tamagawa, K., et al.: Auto-calibration system developed to assimilate amsr-e data into a land surface model for estimating soil moisture and the surface energy budget. Journal of the Meteorological Society of Japan,85A(11), 229-242, 2007.

14. Zhang, G., Zhang, Y., Dong, J., and Xiao, X.: Green-up dates in the Tibetan Plateau have continuously advanced from 1982 to 2011, Proceedings of the National Academy of Sciences, 110, 4309-4314, doi: 10.1073/pnas.1210423110, 2013.

15. Zhou, Y., Guo, D., Qiu, G., and Cheng, G.: Frozen Ground in China, Science Press, Beijing, 450pp, 2000.

16. Zhu, Z., Piao, S., Myneni, R. B., Huang, M., Zeng, Z., Canadell, J. G., Ciais, P., Sitch, S., Friedlingstein, P., and Arneth, A.: Greening of the Earth and its drivers, Nature Climate Change, 6, doi:10.1038/nclimate3004, 2016.

17. Zou, D., Zhao, L., Tonghua, W. U., Xiaodong, W. U., Pang, Q., & Qiao, Y., et al.: Assessing the applicability of modis land surface temperature products in continuous permafrost regions in the central tibetan plateau. Journal of Glaciology & Geocryology, 2015.

---

## Referee Report (RR1)

The manuscript was significantly improved since its last version. I would like to thank authors for addressing all my comments in great details. Below I have several minor editorial comments after which would suggest the manuscript for the publication.

L349. Remove second 'further'.

L 367-378. I felt that there are some redundancies in the paragraph.

L 381 … large scale teleconnections. Add a reference.

L 420 … while no **systematic** relationship is evident with SND. Saying that there is no relationship assumes the independence of SND, when we know that SND significantly affect ground temperatures. Please feel free to choose a better word to finish that sentence.

Figure 4C. Choose different color for -0.008- -0.004 interval

Figure 8 Combine -1.6- -1.2 and < -1.6 intervals into one.

The relationship of SFD with respect to latitude shows nice correlation both for East and West parts of China. I am leaving that up to the authors to consider that plot in the manuscript.

---

## Author Response (AR2)

Dear Dr. Kääb and Dr. Gruber,

   Thank you very much for providing us the opportunity to improve our paper based on the referees' valuable comments. We have revised the manuscript according to your and the reviewers' comments and suggestions. Enclosed please find the revised manuscript, responses to the referees, as well as a list of changes. The responses are marked blue. We hope these revisions have improved our manuscript to make it suitable for publication in "The Cryosphere." If you have any questions or concerns about this paper, please don't hesitate to let me know. We look forward to hearing from you soon. Sincerely yours,

Tingjun Zhang

The summary of the changes and responses to Referees' comments are listed below. The page, line, and figure numbers refer to our revised manuscript. The changes have been indicated in the paper using bold font.

We thank Dr. Kääb for his suggestions of the discussion paper. We have addressed all the comments and made the suggested changes in the revised version of our manuscript. Our point-by-point replies (in blue) to the comments are listed below.

1. Thanks for your detailed response and revisions! I am happy to accept the paper as is, but recommend to include Fig 2, page 19, of the response letter in the paper ("Spatial variability of SFD anomaly for the decades of the 1950s, 1960s, 1970s, 1980s, 1990s, and 2000s, with respect to the 1950–2009 mean across China"). I agree with the referee proposal to include it.
Response: Thanks! We added this figure in the manuscript (figure 8), and changed our figure 7. Meanwhile, we added several sentences in L248-253 (page 9) to describe it.

We thank Dr. Elchin Jafarov for his detailed and insightful review of the revised paper. We have addressed all the comments and made the suggested changes in the revised version of our manuscript. Our point-by-point replies (in blue) to the comments are listed below.

1. L349. Remove second 'further'.
Response: Thanks! We have deleted it.

2. L 367-378. I felt that there are some redundancies in the paragraph.
Response: Thank you for your suggestion. We have carefully edited this paragraph and hopefully removed some of the wordiness and redundancy.

3. L 381 … large scale teleconnections. Add a reference.
Response: We added the reference in L388 (page 14) "(Frauenfeld and Zhang, 2011)".

4. L 420 … while no systematic relationship is evident with SND. Saying that there is no relationship assumes the independence of SND, when we know that SND significantly affect ground temperatures. Please feel free to choose a better word to finish that sentence.
Response: Yes, we agree with you. We revised this sentence in L425 (page 15) "…SND did not show a significant association."

5. Figure 4C. Choose different color for -0.008- -0.004 interval
Response: Nice suggestion. We changed it.

6. Figure 8 Combine -1.6- -1.2 and < -1.6 intervals into one.
Response: Ok, we revised it.

7. The relationship of SFD with respect to latitude shows nice correlation both for East and West parts of China. I am leaving that up to the authors to consider that plot in the manuscript.
Response: Thank you for this suggestion. We have added it as figure 10 with some discussion to the manuscript. Please see L272-280 (page 10).